# Mitogenic and progenitor gene programmes in single pilocytic astrocytoma cells

Zachary J. Reitman[1,2,3,20], Brenton R. Paolella [1,2,20], Guillaume Bergthold[4,5], Kristine Pelton[6], Sarah Becker[6], Robert Jones[6], Claire E. Sinai[7], Hayley Malkin[1], Ying Huang[6], Leslie Grimmet[8], Zachary T. Herbert[8], Yu Sun[1], Jessica L. Weatherbee[1], John A. Alberta[1], John F. Daley[9], Orit Rozenblatt-Rosen[2], Alexandra L. Condurat[2,7,10], Kenin Qian[7], Prasidda Khadka[1], Rosalind A. Segal[1], Daphne Haas-Kogan[11], Mariella G. Filbin[2,7,10], Mario L. Suva [2,12,13], Aviv Regev[2,13,14], Charles D. Stiles[1], Mark W. Kieran[7,10,15,19], Liliana Goumnerova [16], Keith L. Ligon[2,6], Alex K. Shalek[2,17,18], Pratiti Bandopadhayay[2,7,10] & Rameen Beroukhim [1,2,9]

Pilocytic astrocytoma (PA), the most common childhood brain tumor, is a low-grade glioma with a single driver BRAF rearrangement. Here, we perform scRNAseq in six PAs using methods that enabled detection of the rearrangement. When compared to higher-grade gliomas, a strikingly higher proportion of the PA cancer cells exhibit a differentiated, astrocyte-like phenotype. A smaller proportion of cells exhibit a progenitor-like phenotype with evidence of proliferation. These express a mitogen-activated protein kinase (MAPK) programme that was absent from higher-grade gliomas. Immune cells, especially microglia, comprise 40% of all cells in the PAs and account for differences in bulk expression profiles between tumor locations and subtypes. These data indicate that MAPK signaling is restricted to relatively undifferentiated cancer cells in PA, with implications for investigational therapies directed at this pathway.

---

[1] Department of Cancer Biology, Dana Farber Cancer Institute, Boston, MA 02215, USA. [2] Broad Institute of MIT and Harvard, Cambridge, MA 02142, USA. [3] Department of Radiation Oncology, Massachusetts General Hospital, Boston, MA 02215, USA. [4] Hoffmann-La Roche, Product Development, Innovative Pediatric Oncology Drug Discovery, CH-4070 Basel, Switzerland. [5] Centre Hospitalier Universitaire Strasbourg, Service Hématologie-Oncologie Pédiatrique, 67000 Strasbourg, France. [6] Department of Oncologic Pathology, Dana Farber Cancer Institute, Boston, MA 02215, USA. [7] Department of Pediatric Oncology, Dana-Farber Boston Children's Cancer and Blood Disorders Center, Boston, MA 02215, USA. [8] Molecular Biology Core Facilities, Dana-Farber Cancer Institute, Boston, MA 02215, USA. [9] Department of Medical Oncology, Dana-Farber Cancer Institute, Boston, MA 02215, USA. [10] Department of Pediatrics, Harvard Medical School, Boston, MA 02215, USA. [11] Department of Radiation Oncology, Dana-Farber Cancer Institute, Boston, MA 02215, USA. [12] Department of Pathology and Center for Cancer Research, Massachusetts General Hospital and Harvard Medical School, Boston, MA 02114, USA. [13] Klarman Cell Observatory, Broad Institute of Harvard and MIT, Cambridge, MA 02142, USA. [14] Department of Biology, Howard Hughes Medical Institute, Koch Institute, MIT, Cambridge, MA 02139, USA. [15] Bristol-Myers Squibb, Boston, Devens, MA 01434, USA. [16] Department of Neurosurgery, Boston Children's Hospital, Boston, MA 02215, USA. [17] Institute for Medical Engineering and Science, Department of Chemistry, and Koch Institute for Integrative Cancer Research, Massachusetts Institute of Technology, Cambridge, MA 02139, USA. [18] Ragon Institute of Massachusetts General Hospital, Massachusetts Institute of Technology, and Harvard, Cambridge, MA 02139, USA. [19]Present address: Bristol-Myers Squibb, Lawrenceville, NJ 08648, USA. [20]These authors contributed equally: Zachary J. Reitman, Brenton R. Paolella. Correspondence and requests for materials should be addressed to P.B. (email: pratiti_bandopadhayay@dfci.harvard.edu) or to R.B. (email: rameen_beroukhim@dfci.harvard.edu)

Gliomas, cancers composed of cells that resemble glia, are the most frequently lethal primary brain tumors. These tumors are classified as grade I to IV by the World Health Organization based on clinical, genetic, and histopathological criteria. WHO grade II–IV tumors are diffusely infiltrating and generally associated with a poor outcome. Recent efforts have characterized cellular heterogeneity within WHO grade II–IV adult and pediatric gliomas by single cell RNA sequencing (scRNAseq). These studies revealed hierarchical relationships between cancer cells that mimic normal differentiation of brain cells from neural stem cells or glial progenitor cells into mature glia[1–4].

Pilocytic astrocytomas (PAs) are the most common brain tumors in children. These are WHO grade I tumors that can potentially be cured. However, PAs can be associated with considerable treatment-related morbidity from surgical resection, chemotherapy, or radiotherapy[5,6]. PAs that are incompletely resected tend to recur during childhood, but childhood PA patients usually do not succumb to their disease[6–8]. In contrast, higher-grade gliomas are nearly always fatal.

PAs are also distinguished by the simplicity of their genome. Unlike higher-grade gliomas, which usually exhibit multiple driver mutations[9–11], most PAs exhibit a single driver somatic genetic alteration[12–14]. These almost always activate the MAPK pathway, with rearrangements generating the KIAA1549-BRAF fusion oncogene accounting for ~70% of PAs[15]. Targeted therapies directed at the MAPK pathway are undergoing clinical testing for recurrent or incompletely resected PAs[16,17]. However, whether the MAPK pathway is uniformly activated in PA cancer cells remains incompletely understood.

Bulk gene expression analyses of PA and other pediatric low grade gliomas have identified differences in gene expression profiles between PAs that arise in different brain locations[18–21], and also found PA to strongly express immune gene signatures[19,20,22]. It has been unclear whether these differences were due to different contributions of cancer cells and immune cells, or due to different gene programme being expressed in the cancer cells.

Here, we determine the gene expression landscape of PA at single cell resolution. This analysis deconvolutes the contributions of PA cancer and immune cells and indicates heterogenous expression profiles among each of these cell types. PA cancer cells recapitulate a developmental differentiation hierarchy from OPC-like cells to mature astrocyte-like cells. Also, PA cancer cells heterogeneously express MAPK signaling gene programme. Furthermore, PA exhibit a smaller population of proliferative progenitor cells that is more similar to normal glial cells than to the progenitor population of higher grade pediatric astrocytomas. Together, these findings lay a framework for future biologic and therapeutic investigations in PA.

## Results

### Distinguishing pilocytic astrocytoma cells from normal cells.
Recent single-cell RNA-sequencing (scRNA-seq) efforts revealed transcriptional developmental hierarchies across sets of adult gliomas and pediatric histone mutant gliomas. However, how these hierarchies relate to pediatric low-grade gliomas remains unknown. To explore this, we first sought to develop procedures to distinguish pilocytic astrocytoma (PA) cancer cells from intermixed tumor-associated cells. Among high-grade gliomas, expressed somatic mutations and inferred large-scale copy number variations (CNVs) have been previously used to distinguish high-grade glioma cells from tumor-associated cells in scRNA-seq data[1–4]. However, low-grade tumors including PAs often exhibit few or no large-scale CNVs or expressed

mutations[12–14]. We therefore developed a pipeline to identify PA cancer cells using four sources of information: (1) immunolabeling using a PA-specific marker; (2) enhanced BRAF fusion transcript detection; (3) clustering based on RNA-seq profiles; and (4) expressed glial tumor markers (Fig. 1a).

We selected A2B5, a cell-surface marker of glial progenitor cells, as our target for immunolabeling for two reasons. First, A2B5 is known to be enriched in PAs and absent from nonglial cell types[23]. Second, we found that A2B5 detection was enriched six-fold in murine neural stem cells engineered to express KIAA1549-BRAF relative to controls ($p < 0.001$, Student's $t$ test; Supplementary Fig. 1a, b). Representative plots showing our A2B5 gating strategy for viable human PA tumor cells are shown in Supplementary Fig. 1c.

We next combined three approaches to optimize detection of the KIAA1549-BRAF fusion in single cells. First, we used the SMART-seq2 scRNA-seq protocol, which provides full-length transcript coverage, rather than methods that rely on counting 3' transcript ends. Second, we spiked in an oligonucleotide specific for the 3' region of BRAF during cDNA library generation (Fig. 1b and Methods). Third, we performed targeted qPCR for the KIAA1549:BRAF fusion junction in three tumors ($n = 578$ cells). Using SMART-seq2 alone, we found that the fusion was found in no cells or very few cells from these tumors (0–0.7%). Combining the approach with qPCR increased detection rates to 29–54% of all cells. BRAF fusions were only detected in A2B5+ cells (79% of A2B5+ cells, Fisher's exact test, $P < 0.0001$) (Supplementary Fig. 1d, e).

We applied these methods to six PAs to generate scRNA-seq data (Supplementary Data 1). Genetic profiling of bulk tissues confirmed that all tumors contained BRAF alterations, including five tumors with the classic KIAA1549-BRAF 16:9 translocation and one tumor with a noncanonical BRAF duplication event (Supplementary Data 2). We generated scRNAseq data for 1239 cells, of which 931 passed quality control measures and were used for subsequent analyses.

We detected evidence for CNVs in single cells from a subset of PAs. CNVs were inferred from PA cancer cell scRNA-seq data by averaging expression over contiguous stretches of 100 genes[1]. This analysis supported chromosome-arm-level CNVs in four PAs and a silent CNV landscape in two PAs (Supplementary Fig. 2). The inferred CNVs included events previously observed in PA[24] such as gains of chromosomes 5 and 7 in BT646. However, the only inferred CNVs that could be validated by copy number analysis of bulk tissue from the same tumor were a chromosome 7 gain in BT646 and a chromosome 10 gain in BT906 (Supplementary Data 2). We conclude that some CNVs seen in single PA cells could be masked by tumor-associated cells when bulk tissue is analyzed.

### Cancer and noncancer cells separate into distinct groups.
Clustering of PA single cell transcriptomes was highly concordant with A2B5 and BRAF fusion status. Visualization of the transcriptomic profiles using nonlinear dimensionality reduction (t-SNE) revealed substantial overlap between cells from each tumor, suggesting shared transcriptomic patterns across all tumors (Fig. 2a). Across all tumors, A2B5 positive (A2B5+) cells clustered together (Fig. 2b), and these tended to be the BRAF+ cells (Fig. 2c). These two groups (A2B5+, BRAF+ and A2B5−, BRAF−) defined the first principle component in principle component analysis (PCA) of the same data (Supplementary Fig. 3a-d). Shared nearest neighbors and nonnegative matrix factorization were used as parallel approaches to identify clusters of cells (Methods). These approaches revealed five clusters (Fig. 2d and Supplementary Fig. 4a-c). Two clusters were almost

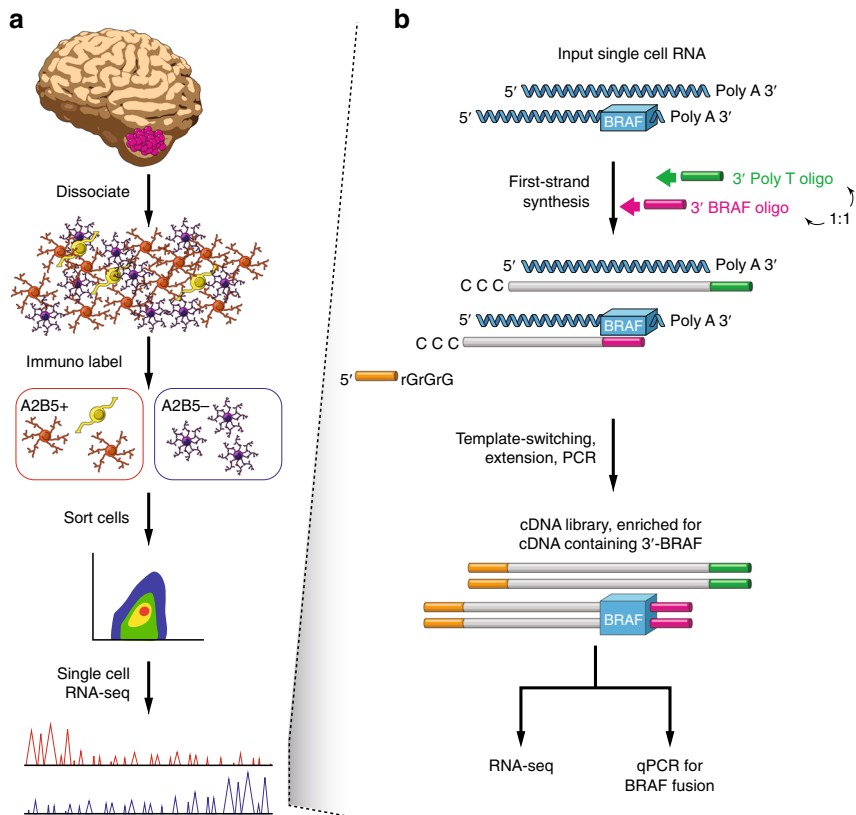

**Fig. 1** Process for single cell RNA-seq and cancer cell identification in low grade brain tumors. **a** Workflow for rapid dissociation, A2B5 immunolabeling and flow sorting to enrich for A2B5+ glial progenitor cells, followed by single cell RNA sequencing. **b** Schematic for BRAF Spike-in-Seq. A 3′ BRAF oligonucleotide (oligo) is added in 1:1 stochiometry with a general 3′ poly-T oligonucleotide to enhance representation of KIAA1549:BRAF in first-strand synthesis pool. First-strand synthesized nucleic acids are then subjected to whole transcriptome amplification, tagmentation, and next-generation sequencing for RNA-seq analysis. Quantitative PCR on the cDNA library is directed at the KIAA1549-BRAF fusion junction to maximize detection of this transcript

entirely comprised of A2B5+, BRAF fusion-containing cells (clusters 0 and 1) (Fig. 2b–d). The other three clusters were almost exclusively A2B5-containing, non-fusion-containing cells (clusters 2 to 4). Accordingly, clusters 0 and 1 expressed glial markers associated with PA such as *OLIG2*, *APOD*, and *PDGFRA* (Fig. 2e). Clusters 2 to 4 expressed markers associated with immune cells, such as *CCL3*, *SOD2*, and *IL32* (Fig. 2f).

**Immune cells contribute to bulk PA expression profiles**. PA have been previously reported to exhibit heterogeneity with the presence of immune cells intertwined with cancer cells[19,20,22]. Indeed, we also found evidence of immune cells within our scRNAseq data (Fig. 3a, b). We examined tumor-associated cells for gene signatures of microglia and macrophages collected throughout the mouse lifespan in states of health and disease[25] (Supplementary Fig. 5a and Supplementary Data 3). All tumors contained cells belonging to a cluster exhibiting expression of microglia-related genes (e.g., *CCL3*, *CCL2*, *C3*, *CD74*, *HLA-DRA*) that scored most highly for microglia from healthy adult mice (Fig. 3b and Supplementary Fig. 5b). These comprised approximately 30% of all single cells within individual tumors and were the most abundant immune cells within PA. We also identified clusters of immune cells exhibiting markers of macrophage differentiation in two tumors (e.g., *SOD2*, *S100A9*, *CXCL8*, and *IL1R2*; Supplementary Fig. 5c) and of T lymphocytes in four tumors (e.g., *IL32*, *GZMA*, *GZMK*, *CD52*, and T cell receptor genes).

Recognition of these cell types enabled us to deconvolute bulk tumor profiles. We estimated contributions of each cell type signature to the bulk tumor transcriptional data from 151 pediatric low grade gliomas[19] (PLGGs, Fig. 3c). Similar to our single cell profiles, microglia contributed a median of 30% (IQR: 20–44%) of the bulk transcriptomic signature (Fig. 3d). In contrast, cancer cells contributed slightly more than half of the total bulk transcriptional signal on average (median 57%, IQR: 48–66%). Our prior analyses of bulk transcriptional data have found a strong distinction between profiles from supra-tentorial and infra-tentorial PLGGs[19]. We find that the genes specifically expressed in the supratentorial tumors from these publications (such as *HLA-DRA*, *LYZ*, *CD74*, *F13A1*, and *TRIM22*, Supplementary Fig. 6a, b) tend to be expressed in cells within the microglia-type cluster, whereas the infratentorial tumors have higher expression of cancer cell-related genes (Fig. 3e). This effect was also seen for genes that were differentially expressed in supratentorial vs. infratentorial tumors in an independent study[20] (Supplementary Fig. 6c–e). We conclude that the ratio between microglia and cancer cells is a major driver of differences in bulk transcriptional profiles between supra-tentorial and infra-tentorial PLGGs.

Prior reports of immune cells within PLGGs have raised the possibility of immunotherapy for children with PA[26–28], however our data suggest lack of expression of relevant immune checkpoint signaling genes that can be targeted with current clinically relevant inhibitors. In adult populations, the efficacy of emerging immune checkpoint blockade therapies (such as PD1

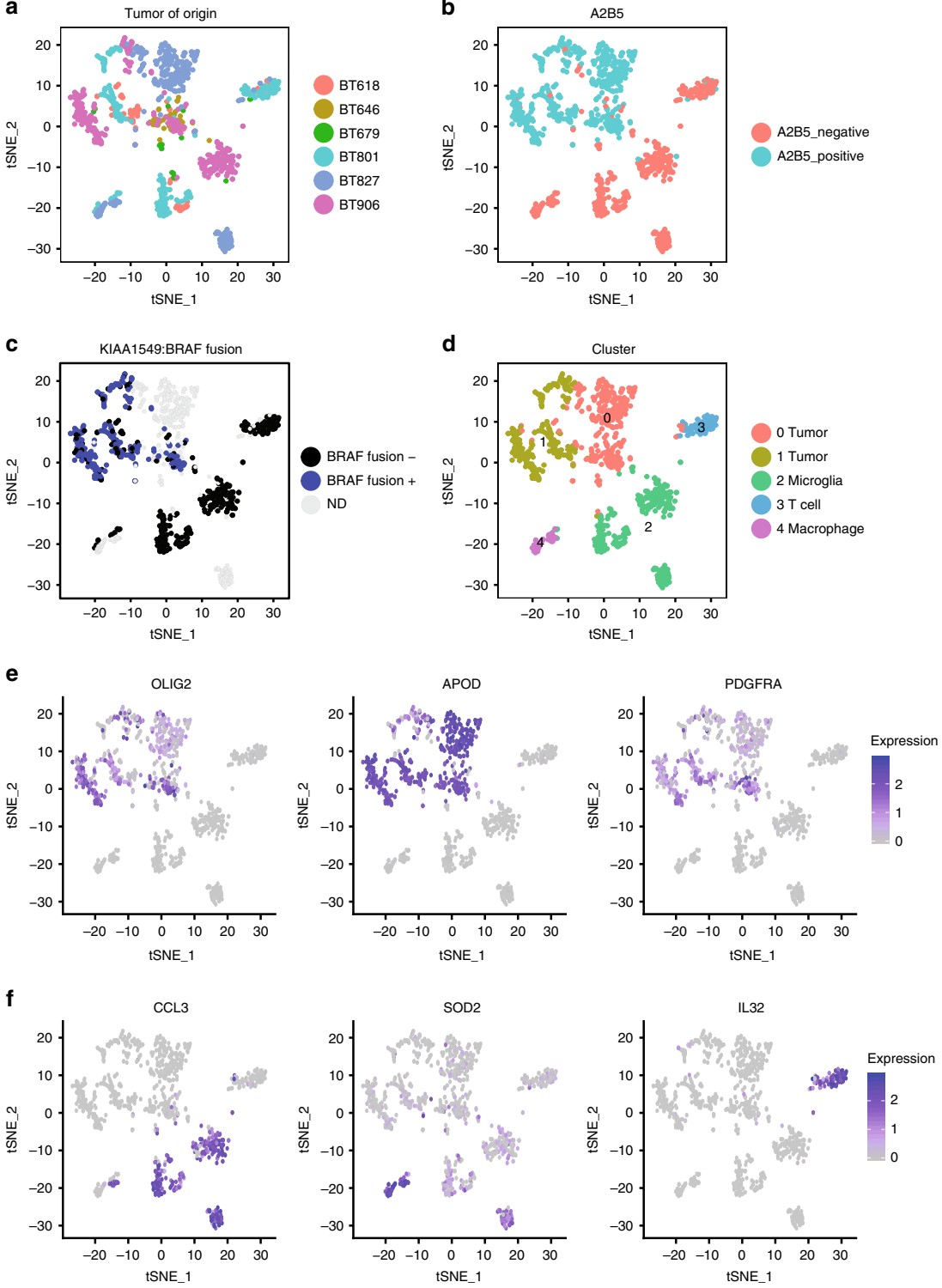

**Fig. 2** Clustering of transcriptomic profiles corresponds to A2B5 and KIAA1549:BRAF status of PA cells. **a** t-SNE plot showing PA single cells colored by six tumors of origin. **b** t-SNE plot showing PA single cells colored by A2B5 glial progenitor marker status as determined by immunolabeling. **c** t-SNE plot showing PA single cells colored by KIAA1549:BRAF status as determined by BRAF Spike-in-Seq for cells undergoing quantitative PCR directed at the KIAA1549:BRAF fusion junction. **d** t-SNE plot showing PA single cells colored by shared nearest neighbors clustering of transcriptomic profiles revealing tumor clusters (0 and 1), a microglia cluster (2), a T cell cluster (3), and a macrophage cluster (4). **e** Relative expression of glial markers associated with PA *OLIG2*, *APOD*, and *PDGFRA*. Scale shows log-normalized read counts. **f** Relative expression of markers of immune cells including *CCL3* for microglia, *SOD2* for macrophages, and *IL32* for T cells. Scale shows log-normalized read counts

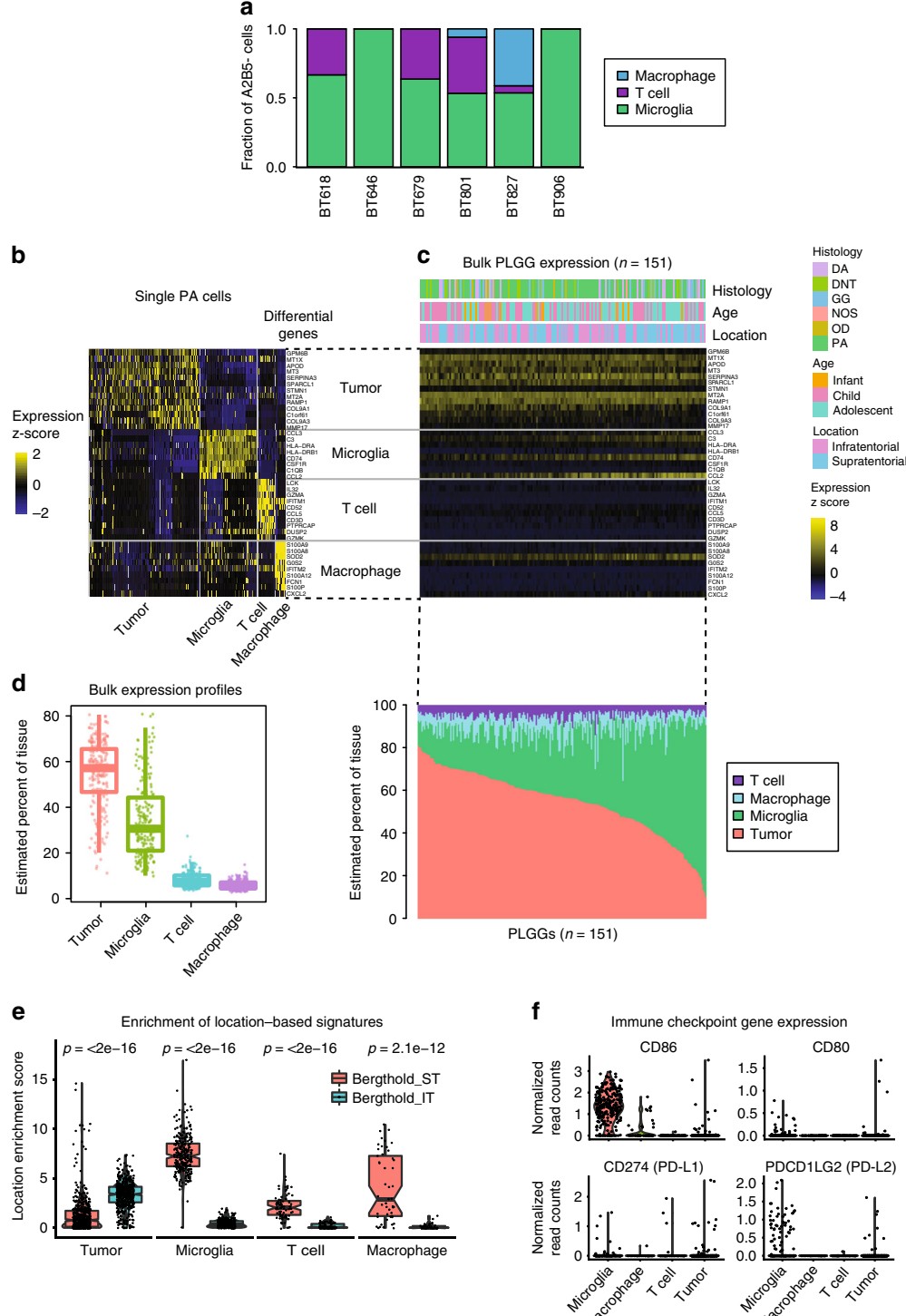

**Fig. 3** Single cell RNA-seq analysis of immune cells in PA. **a** Fraction of microglia, macrophages, and T cells identified among A2B5− cells in six tumors subjected to scRNA-seq. **b** Heat map showing top differentially expressed genes (rows) for PA immune cell and cancer cell clusters. Cells (n = 931, columns) are arranged by cancer cell and immune cell clusters. **c** Expression of the differential genes from panel b in bulk expression profiles of 151 pediatric low grade gliomas[19] (PLGGs). Histological subtypes are designated as: DA diffuse astrocytoma, DNT dysembryoplastic neuroepithelial tumor, GG ganglioglioma, OD oligodendroglioma, NOS not otherwise specified. **d** Estimated proportion of tumor and immune cells in 151 pediatric low grade gliomas[19], based on relative expression of gene signatures for tumor and immune cell types derived from single cell RNA-seq analysis. Center line shows median, hinges show 1st through 3rd quartiles, and whiskers extend to 1.5 times the interquartile range or the value fathest from hinge, whichever is less. **e** Enrichment of gene signatures derived from comparison between bulk expression profiles of supratentorial and with infratentorial tumors[19], among cell types in the single-cell dataset. P-values are for Wilcoxon rank sum test. **f** Expression of immune checkpoint ligand genes CD86, CD80, CD274 encoding PD-L1, and PDCD1LG2 encoding PD-L2 in tumor and immune compartments of six PA tumors analyzed with scRNA-seq. Center line shows median, hinges show 1st through 3rd quartiles, and whiskers extend to minimum and maximum data points

inhibitors) has been linked to expression of immune checkpoint signaling molecules in specific cancer and immune cell compartments[29]. However, immune checkpoint blockade has not yet demonstrated efficacy for patients with primary brain tumors or for pediatric patients. To prioritize investigation of immune checkpoint pathways to target in select PA patients, we first examined expression of ligands for PD1 and CTLA4 in PA tumor and immune cells (Fig. 3f). Ligands for PD1 (*CD274* encoding PD-L1 and *PDCD1LG2* encoding PD-L2) were minimally expressed in cancer cells. This is consistent with observed low rates of PA cancer cell PD-L2 staining observed by immunohistochemistry[26,28]. Similarly, ligands for CTLA4 (*CD86* and *CD80*) were almost never expressed in cancer cells. *CD86*, encoding a ligand of CTLA4, exhibited robust expression in microglia, as is often seen in antigen presenting cells, but the other ligands were seldom expressed in any of the immune cells in our dataset. We next examined additional components of checkpoint signaling pathways, including MHC I expression and expression of ligands that interact with immune checkpoint receptors other than PD1/CTLA4 (Supplementary Fig. 7a, b). Robust expression of MHC class I genes (HLA-A, HLA-B, and HLA-C) was seen in PA tumor and immune cells, suggesting that antigen presentation is intact. Expression of checkpoint receptors and ligands varied between tumors and cell types, but in general the highest levels of expression in microglia were seen for ICOSLG (ligand of ICOS receptor), CD276 (B7-H3), TNFRSF14 (ligand of BTLA), and LGALS9 (ligand of TIM3). These analyses suggest that the roles of these receptors and ligands deserve further study in PA.

**Single PA cells resemble oligodendrocyte precursor cells.** Having mapped the architecture of immune cells within the tumors, we next sought to identify genes that were specifically expressed in PA cancer cells. To do so, we compared the two A2B5+, BRAF fusion-associated clusters to the tumor-associated cells in the other three clusters (Fig. 4a, b and Supplementary Data 4). The A2B5+, BRAF+ population was marked by high levels of canonical markers of PA cancer cells, including oligodendrocyte-associated markers (OLIG1, OLIG2, PDGFRA), glial markers (GFAP, APOD, APOE), serine proteases (SERPINA3, SERPINE2), and other PA markers such as PLEKHB1. At pathway level, gene set enrichment analysis (GSEA) revealed enrichment of gene signatures associated with CNS tumors ($n = 16$) or normal CNS cell types such as neurons, oligodendrocytes, or astrocytes ($n = 15$) (Supplementary Data 5).

PA cancer cells exhibited enrichment of oligodendrocyte precursor cell (OPC) signatures relative to normal cells. To identify the normal brain cell types that PA cancer cells resemble, we compared our data to human single cell transcriptomic atlases. The PA cancer cell clusters were most enriched for OPC gene signatures from a single cell atlas of the developing midbrain[30] as opposed to other types of mature or developing neurons, radial glia, or mature glia (Fig. 4c and Supplementary Fig. 8a). We found OPC signatures derived from additional independent normal brain single cell atlases (adult cortex[31] and developing cortex[32]) to also be the most highly enriched within the PA cancer cells (Supplementary Fig. 8b, c). Furthermore, we observed enrichment of OPC-like signatures derived from subpopulations of high grade gliomas (Supplementary Fig. 8d, e). These analyses indicate that PA cancer cells most closely resemble OPCs.

PA cancer cells were dissimilar to those obtained from higher-grade gliomas. We compared PA tumor scRNAseq data to data from H3K27M-mutated midline high grade gliomas[4] and from IDH-mutated astrocytomas and oligodendrogliomas of intermediate grade[3]. While all tumors exhibited signatures of OPCs

and radial glia, neural stem cell and neuroblast signatures were more enriched in the higher-grade tumors (Fig. 4d) and PA scored more strongly for OPC gene signatures (Fig. 4e and Supplementary Fig. 9a,b). These comparisons to normal brain single cell gene signatures revealed that PA cancer cells on average resemble a normal OPC that may be more committed than the cancer cells in higher-grade gliomas.

**MAPK signaling and glia-like programme in distinct PA cells.** We next used unbiased analyses to identify gene programme expressed in subpopulations within the PA cancer cells. In a principal component analysis, PC1 distinguished cells characterized by either transcription factors expressed downstream of MAP kinase signaling cascades (e.g., JUN, FOSB, EGR1) or genes expressed in mature astrocytes (e.g., B2M, APOD, Fig. 5a and Supplementary Data 6). PC2 distinguished cells characterized by oligodendroglial markers (e.g., ZAK and QKI) (Fig. 5b). We found that the MAPK signaling and astrocyte-like signatures were expressed by large numbers of cells, whereas only a few cells exhibited the oligodendrocyte-like signature (Fig. 5c). Most cells expressed a combination of the MAPK and astrocyte-like signatures. We next nominated a list of glial and progenitor markers and transcription factors for analysis. We found that while almost all cancer cells expressed OPC-associated signal transduction factors OLIG1, OLIG2, and PDGFRA, expression of progenitor-associated transcription factors such as SOX2 and SOX10 were highly expressed in cells expressing the MAPK programme (Fig. 5d). We also found that BRAF normalized read counts were higher in cancer cells expressing the MAPK gene programme (tumor cluster 0) compared to those expressing the AC-like gene programme (tumor cluster 1) (Supplementary Fig. 10, $P = 0.003$ using Fisher's method). This analysis indicated that PA cancer cells comprise a developmental spectrum that extends from a minority of cells that express high levels of the MAPK signaling programme and little AC-like or OC-like differentiation, to a much larger group of AC-like cells with lower levels of MAPK signaling (Fig. 5e). In addition, there are a few cells that express OC-like signatures, and these have relatively high MAPK signaling scores.

We validated expression of specific genes in PA using orthogonal approaches. We used fluorescence immunohistochemistry (F-IHC) to examine expression of OLIG2, a progenitor marker that was expressed in a majority of PA cancer cells based on scRNA-seq, and of GFAP, a top marker of the AC-like gene programme, in the context of the PA tumor architecture (Fig. 6a, b). GFAP expression was spatially confined to compartments of fibrillary cells (Fig. 6c, d). Olig2 expression was observed in the majority of tumor cells, and was expressed highly in microcystic areas of the tumor and also in the fibrillated areas (Fig. 6e, f). Ki-67, a marker of cellular division, was expressed rarely in up to ~2% of cells (Fig. 6g, h). Tumor cells expressing Olig2, GFAP, or both by F-IHC were observed in all tumors analyzed (Fig. 6i). Also, JUN and APOD, the top markers of the MAPK signaling and AC-like gene programme, were highly expressed in archived PA tissues using RNA in situ hybridization (RNA ish, Fig. 6j). Similar to the scRNA-seq data, we observed cells that expressed only JUN, only APOD, or demonstrated intermediate expression of both markers. These F-IHC and RNA ish data validate compartment-specific expression of specific transcripts seen by scRNA-seq.

To examine whether a developmental process is plausible based on our scRNA-seq data, we used a reverse graph embedding approach[33] to construct a cellular trajectory from the cancer cell RNA-seq profiles. This analysis ordered cells within the trajectory in pseudotime. In this analysis cells with low pseudotime values

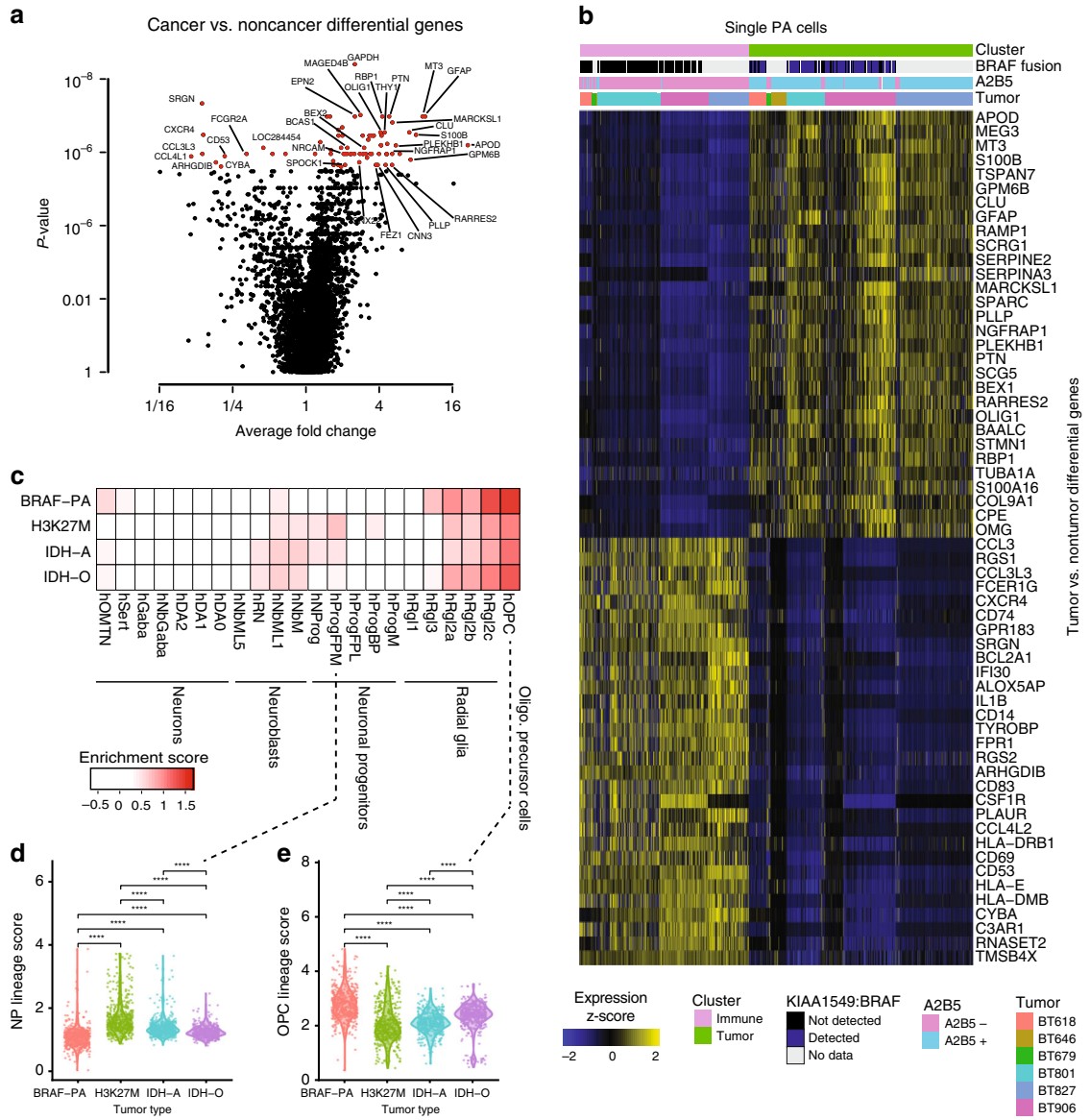

**Fig. 4** Genes and normal brain gene sets expressed by PA cancer cells. **a** Volcano plot showing differentially expressed genes between tumor and non-tumor clusters. P-values are Wilcoxon rank sum test P-values. Genes in red are FDR < 0.05 with Bonferroni correction. **b** Heat map showing expression z-scores for top differentially expressed genes between cancer cells and tumor-associated cells (n = 931) in PA. **c** Heat map showing mean enrichment scores for normal developing midbrain gene signatures[30] for single cells from PA (BRAF-PA), H3K27M-mutated pediatric high-grade midline gliomas[4] (H3K27M), IDH-mutated astrocytomas[3] (IDH-A), and adult IDH-mutated oligodendrogliomas[2] (IDH-O). OMTN oculomotor and trochlear nucleus, Sert serotonergic, NbM medial neuroblast, NbDA neuroblast dopaminergic, DA0-2 dopaminergic neurons, RN red nucleus, Gaba1-2 GABAergic neurons, mNbL1-2 lateral neuroblasts, NbML1-5 mediolateral neuroblasts, NProg neuronal progenitor, Prog progenitor medial floorplate (FPM), lateral floorplate (FPL), midline (M), basal plate (BP); Rgl1-3 radial glia-like cells, Mgl microglia, Endo endothelial cells, Peric pericytes, Epend ependymal, OPC oligodendrocyte precursor cells. **d** Violin plot showing ProgFPM lineage scores for cancer cells of each tumor type. **e** Violin plot showing OPC lineage scores for cancer cells of each tumor type. ****P < 0.0001 (Wilcoxon rank sum test)

occupy early states in the inferred developmental process and cells with high pseudotime values occupy later states (Fig. 7a). Cells from all tumors were well-represented throughout the trajectory (Fig. 7b). The initial low-pseudotime cells had high MAPK signaling scores and highly expressed MAPK genes (Fig. 7c, Supplementary Fig. 11a). In contrast, high-pseudotime cells had increasing expression of AC-like scores and of AC-like genes (Fig. 7d, Supplementary Fig. 11b). Most cells did not score highly for the OC-like gene programme, but the few that did were mostly in a distal branch with high pseudotime values (Supplementary Fig. 11c, d). These results support a developmental process in PA cancer cells.

Expression of activated BRAF can paradoxically lead to oncogene-induced senescence in vitro, which may explain the relatively favorable clinical outcomes associated with PA[34,35]. We sought to determine which PA cells were proliferative and which were senescent within the transcriptional population structure that we identified within PA. To do so, we examined whether senescence[36] or cell cycling[37] gene programme were correlated with either the MAPK signaling or mature glia gene programme among cancer cells in our dataset (Fig. 8a, b). We found that the MAPK and senescence programme were the most strongly correlated pair in this analysis (Spearman $\rho = 0.24$, Bonferroni-adjusted $q < 10^{-6}$). The MAPK gene programme score was also

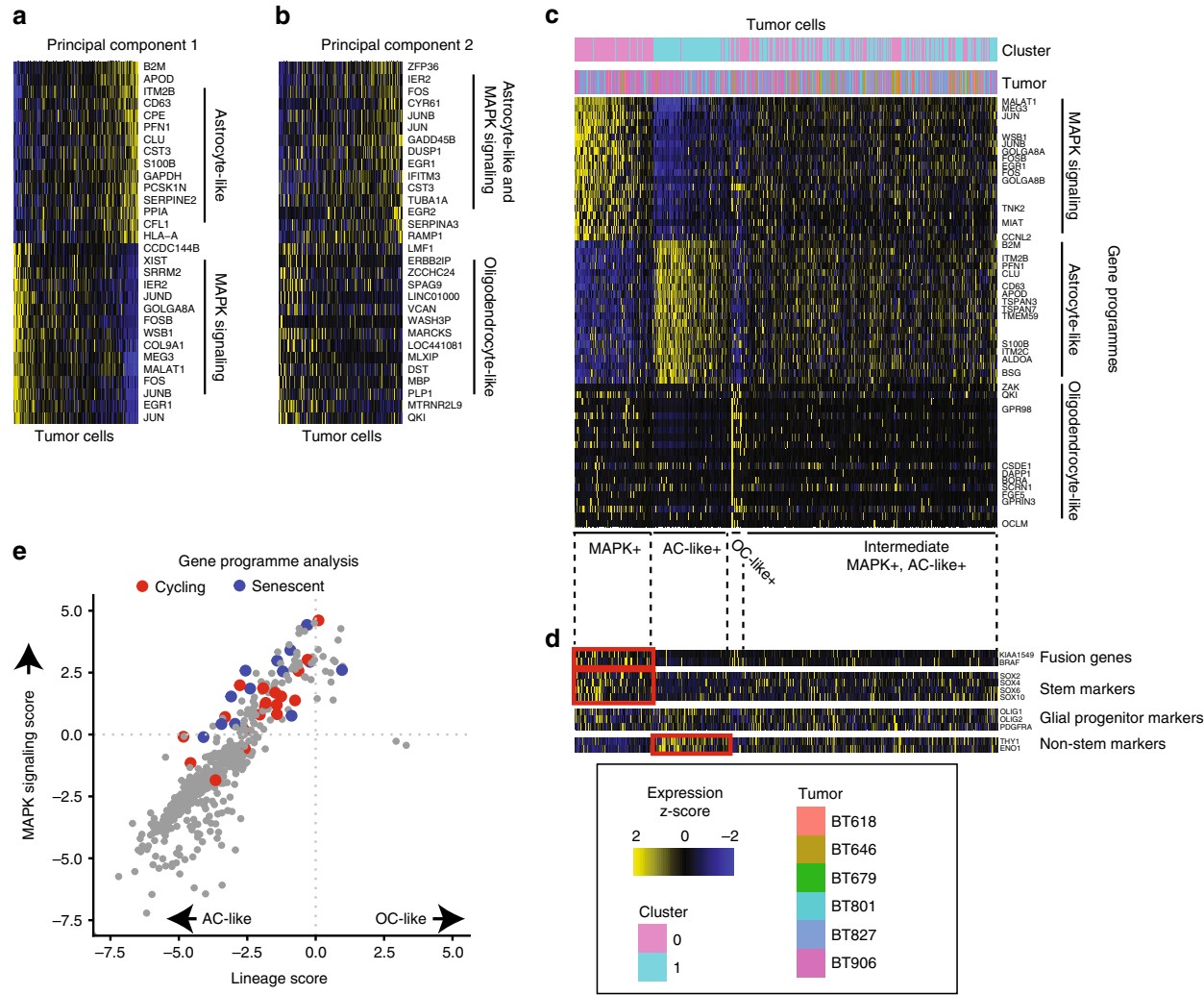

**Fig. 5** MAPK signaling and glia-like gene programme in distinct subsets of PA cells. **a** Cancer cells ($n = 531$, columns) ranked by principal component 1 derived from PCA of cancer cells. Rows indicate the 15 top and 15 bottom genes associated with principal component 1. **b** Cancer cells (columns) ranked by principal component 2; rows indicate genes associated with principal component 2. **c** Heat map showing expression of top 20 genes (rows) from MAPK signaling, astrocyte-like, and oligodendrocyte-like gene programme, across cancer cells (columns) ranked by their expression of each programme. **d** Expression of selected tumor markers in PA cancer cells from panel c. Red boxes mark populations with >2-fold increased mean scaled expression ($P < 0.0001$ non-parametric Wilcoxon rank sum test) of the indicated genes compared to all other cells in the dataset. **e** Relative expression of MAPK signaling and oligodendrocyte-like (OC-like) or astrocyte-like (AC-like) programme in all PA cancer cells. Cells expressing a cell cycling programme are shown in red. Cells expressing a senescence signature are in blue

significantly correlated with BRAF expression ($\rho = 0.14$, $q = 0.008$) and with the cell cycling score ($\rho = 0.12$, $q = 0.04$). Cells expressing high levels of MAPK signaling and low AC-like or OC-like scores were enriched for cells expressing markers of the cell cycle (Fig. 5e). As expected, the senescence and cell cycling scores were not correlated ($\rho = 0.015$, NS). These results demonstrate that proliferating and senescent cancer cells are mutually exclusive subpopulations within a compartment of PA cancer cells that expresses a MAPK gene programme. Together, these findings link a MAPK transcriptomic programme with senescence in vivo.

Since we observed differential expression of oncogenic BRAF in cancer cells with different cell states, we hypothesized that expression of oncogenic BRAF may influence the state of PA cancer cells by modulating one or more of the gene programme identified above. To test this hypothesis, we examined single cell transcriptomes of mouse neural stem cells (mNSCs) that we engineered to express KIAA1549-BRAF fusion[38]. In parallel, we

examined mNSCs expressing BRAF-V600E, another common BRAF alteration found in pediatric gliomas[38], or vector only control. Single cell RNA-seq data were generated for 487 mNSCs after quality filtering ($n = 170$ vector control mNSCs, $n = 154$ KIAA1549-BRAF mNSCs, and $n = 163$ BRAF-V600E mNSCs). mNSCs expressing each construct clustered separately from each other (Fig. 9a). The top differentially expressed genes for mNSCs expressing each construct were identified (Fig. 9b).

Comparison of differentially expressed gene lists in the C2 MSigDB database revealed that mNSCs expressing oncogenic BRAF constructs has upregulation of genes in high-CpG-density promoters containing histone H3 dimethylation at K4 and trimethylation at K27 (H3K27M) in brain[39]. In contrast, genes differentially expressed in the vector control cells significantly overlapped with a proneural glioma signature[40]. To determine whether any of the constructs may upregulate or downregulate gene programme identified in our human PA single cell RNA-seq analysis, we computed overlaps between the list of genes

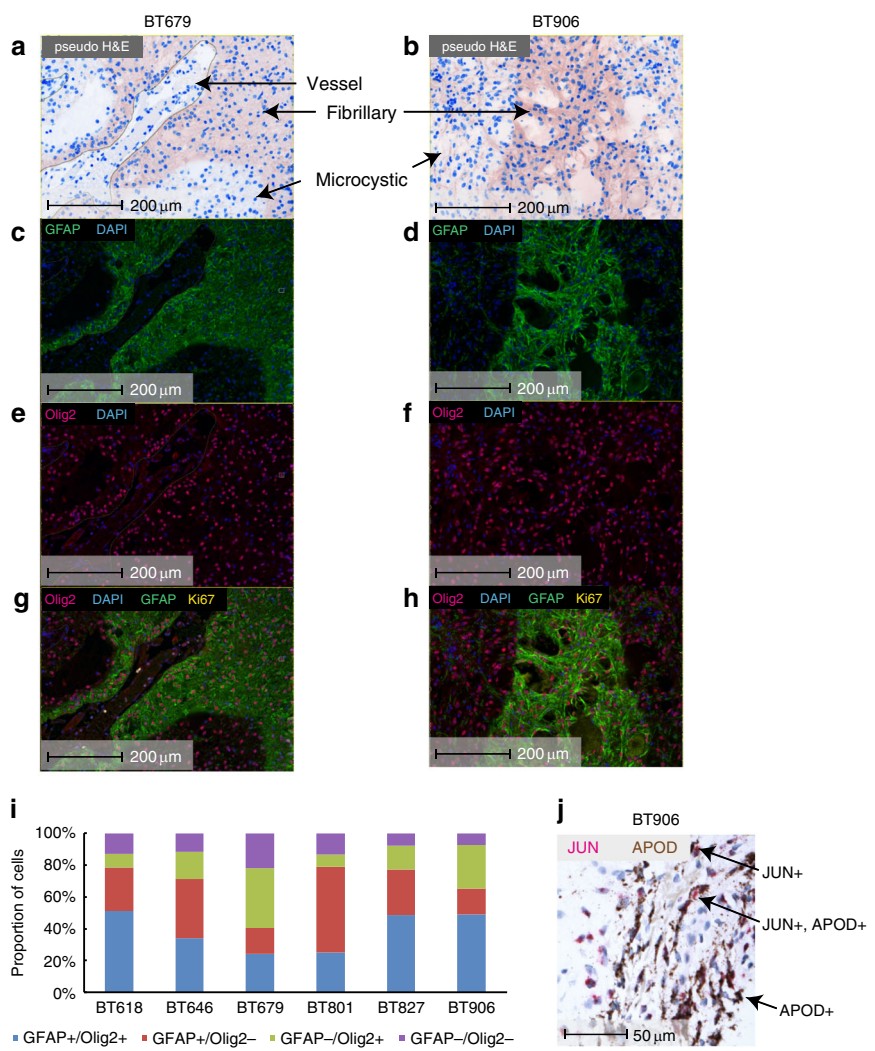

**Fig. 6** Compartment-specific expression of PA genes. Representative images are shown for F-IHC analysis of BT679 (**a**, **c**, **e**, **g**) and BT906 (**b**, **d**, **f**, **h**). **a–b** Pseudo H&E staining. Microcystic and fibrillary components of PA biphasic histology are called out by arrows for both tumors. A vessel is called out as a negative control for tumor cells in BT906. **c–d** GFAP F-IHC, with GFAP positivity indicated in green. **e–f** Olig2 F-IHC is shown with Olig2 positivity indicated in red. **g–h** Overlay of Olig2 and GFAP F-IHC. Rare Ki67 positive cells are also shown in yellow. A blood vessel is outlined in the bottom left of the BT906 panels (**b**, **d**, **f**, **h**). **i** Quantification of Olig2 + and of GFAP + cells for six tumors. $n = 14158$ cells were analyzed for BT618; $n = 22443$ for BT646; $n = 17234$ for BT679; $n = 5678$ for BT801; $n = 12674$ for BT827; $n = 12696$ for BT906. **j** Representative image of RNA in situ hybridization of PA formalin-fixed paraffin-embedded tissue from BT906 showing expression of a top gene in the MAPK signaling gene programme (*JUN*) and of a top gene in the astrocyte-like gene programme (*APOD*). Source data are provided as a Source Data file

differentially upregulated in mNSCs expressing each construct and the PA-derived gene programme (Fig. 9c). Only the AC-like programme and the genes upregulated in the vector control cells (and thus downregulated in the KIAA1549-BRAF and BRAF-V600E cells) demonstrated a significant overlap (Fig. 9d, Bonferroni-adjusted Fisher's exact test $q = 0.002$). Indeed, AC-like gene programme scores were significantly lower for mNSCs expressing the BRAF constructs compared to controls (Fig. 9e). This analysis demonstrates that oncogenic BRAF expression can oppose expression of mature glia gene programme, indicating that dynamics of BRAF expression may contribute to heterogeneity in cancer cell states found in PAs.

**Distinct and shared programme in PA and high-grade gliomas.** We next sought to determine whether gene programme expressed in subsets of PA cells are similar to the gene programme identified in higher-grade gliomas. We compared the PA-derived MAPK signaling, AC-like, and OC-like gene programme to five

gene programme derived from pediatric H3K27M midline gliomas: stem/cell cycle, OPC variable, OPC shared, AC-like, and OC-like. The AC-like mature glia gene programme derived from PAs and H3K27M high-grade gliomas exhibited significant similarities. First, there was significant overlap between the genes in the two AC-like gene programme (Fisher's exact test $P < 0.00001$), with shared genes including *B2M*, *APOE*, *CLU*, *SPARC*, *HLA-B*, and *HLA-C* (Fig. 10a). Nevertheless, the PA AC-like programme did exhibit several differences from the K27M AC-like programme, including higher levels of *APOD* and *GAPDH* genes in the PA programme and *HEY1* in the K27M programme. Second, the PA and H3K27M-derived AC-like gene programme were highly correlated with each other among AC-like cells derived from PA, H3K27M gliomas, or IDH-mutated gliomas (Spearman's $\rho > 0.45$, $P < 10^{-9}$ for each, Fig. 10b and Supplementary Fig. 12a-e). In contrast, oligodendrocyte-like gene signatures from PA and high-grade pediatric gliomas revealed little overlap aside from an oligodendrocyte-associated myelinating

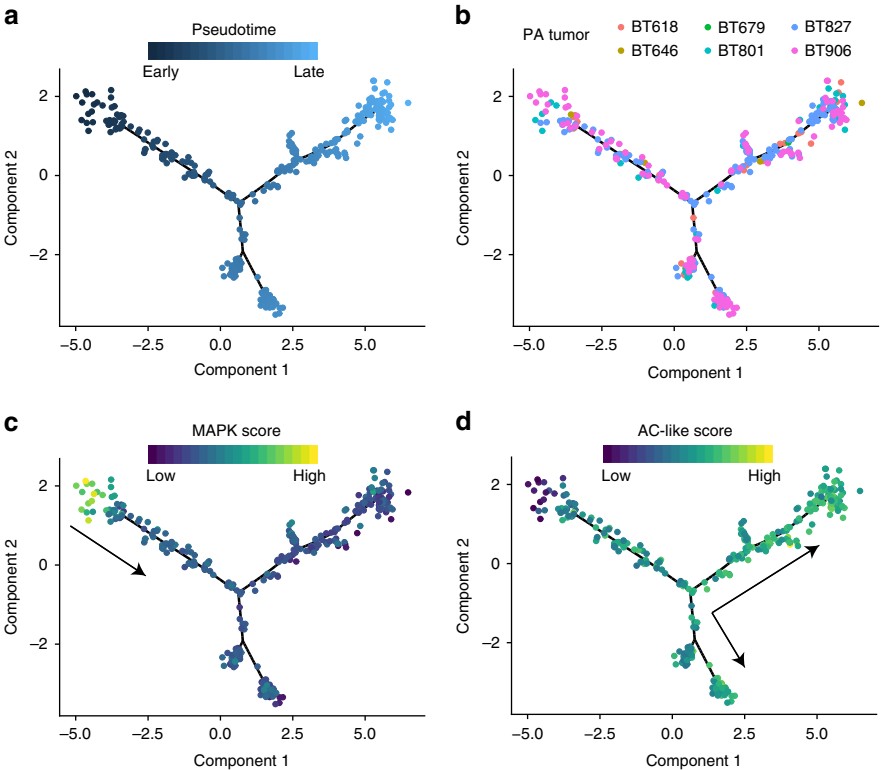

**Fig. 7** Inferred trajectories between PA cancer cells. PA cancer cells are plotted based on a lineage trajectory inferred from RNA-seq data. **a** Cancer cells are colored based on pseudotime inferred from cell trajectory, with pseudotime beginning in the branch on the top right and advancing as cells approach the left and bottom branches. **b** Cells are colored by tumor of origin. **c** Cells are colored by MAPK signaling gene programme score. **d** Cells are colored by AC-like gene programme score

factor *BCAS1*[41] (Supplementary Fig. 12f). This analysis revealed shared features between subpopulations of PA cells and higher-grade gliomas that resemble mature astrocytes.

In contrast to the AC-like signatures, the PA-derived MAPK gene programme was not similar to gene programme expressed in specific compartments of higher-grade gliomas. Expression of the MAPK signaling gene programme was correlated with an OPC-shared gene programme derived from H3K27M gliomas that is generally expressed in H3K27M high-grade subpopulations (Supplementary Fig. 12g-k). However, expression of the MAPK signaling gene programme expression was not correlated with expression of any of the H3K27M glioma-derived gene programme that were expressed in specific tumor subpopulations. For instance, although in PAs the MAPK signaling gene programme was associated with cycling cells that lack markers of differentiated cells, none of the 50 genes from the PA-derived MAPK signaling gene programme overlap with the 50 genes from the H3K27M stem/cell cycling signature (Fig. 10c). Furthermore, the MAPK signaling genes were not highly expressed in the H3K27M stem cell population, nor were H3K27M stem/cell cycle genes highly expressed in PA MAPK cells—indeed, we observed significant anticorrelations between the two programme in cells from both tumor types (Fig. 10d). This distinction was driven by higher expression of *JUN*, *JUNB*, *FOS*, *FOSB*, etc., in the PA cells and by near-absence of neuronal stem cell associated genes such as *TUBA1B* in PA. These analyses show that the MAPK signaling programme is distinct from stem programme in a higher-grade glioma.

We found that PA cancer cells expressing the MAPK signaling gene programme and the PA cancer cells expressing the AC-like gene programme both resembled more committed glial

progenitors compared to higher-grade gliomas. To examine the developing brain cells that the PA cell subpopulations most resemble, we calculated enrichment scores for developing human midbrain cell types[30] for PA cells expressing either gene programme, and for the stem and mature glia-like populations found in H3K27M and IDH-mutated gliomas. This analysis revealed that the PA cells expressing either the AC-like or the MAPK signaling gene programme both most resembled OPCs more than earlier neuronal progenitors (Fig. 10e). Both of these PA subpopulations clustered with the AC-like cancer cells from IDH-mutated and H3K27M gliomas based on their developing brain cell type enrichment scores. The stem, OC-like, and OPC-like populations from the higher-grade gliomas exhibited enrichment for neural progenitor cell signatures that was absent from the PA cancer cells and from the AC-like cells from the higher-grade gliomas. Together, these results show that PA and higher-grade gliomas all contain subpopulations that express partially overlapping mature glia-like gene programme, and that PA also contains a population of cells expressing a MAPK signaling programme that resembles a more committed glial progenitor compared to the stem and progenitor populations from higher-grade gliomas (Fig. 10f).

## Discussion

Here we examine single cell transcriptional profiles of pediatric low-grade brain tumors using a workflow developed specifically to detect KIAA1549-BRAF rearranged PA cells. We found that a considerable portion of bulk PA gene signatures are contributed by nonmalignant microglia. Peeling away these nontumor gene signatures, we noticed that most PA cancer cells expressed a gene

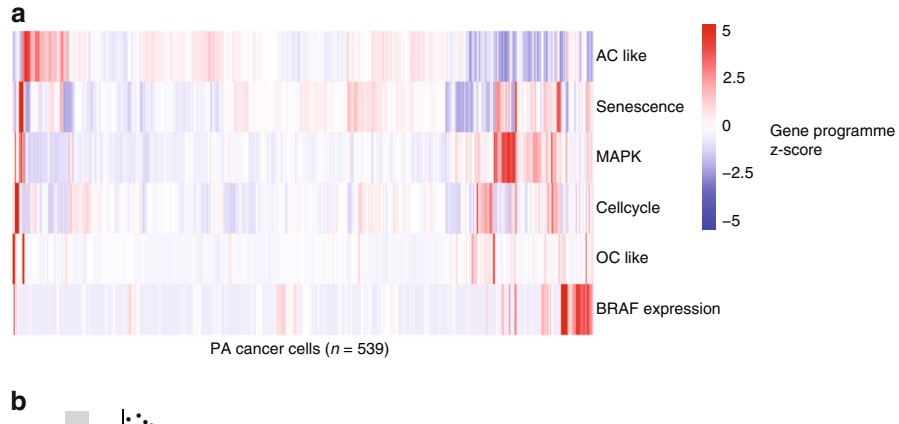

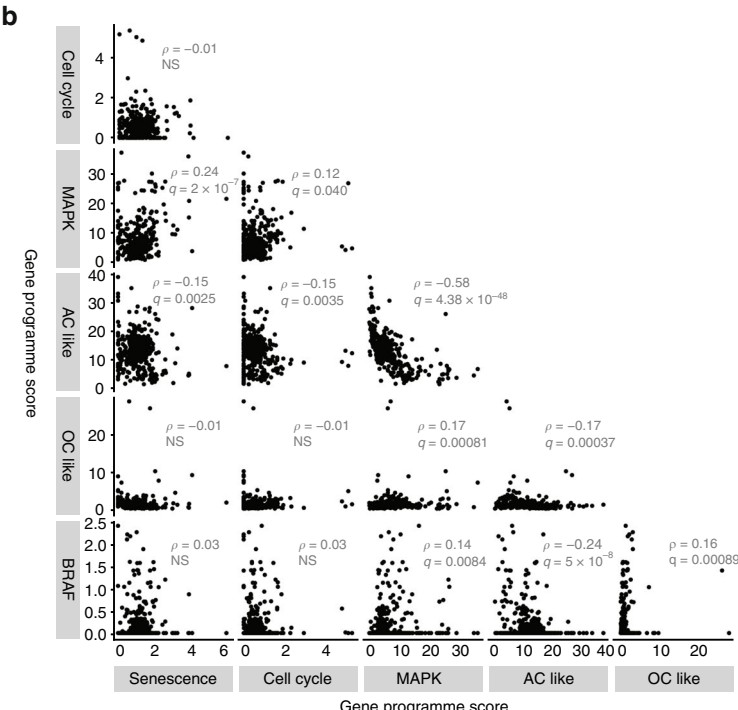

**Fig. 8** Relationship between senescence, cell cycling, MAPK, and glia-like programme in PA cancer cells. **a** Heat map showing ($n = 539$) PA cancer cells as columns. Rows show lineage score, MAPK signaling gene programme score, cell cycle score, or senescence score, or BRAF expression. Data are row-scaled and arranged by unsupervised hierarchical clustering. **b** Plots showing indicated signature scores for PA cancer cells ($n = 539$). Each panel on the bottom left shows a pair of signature scores and/or BRAF expression values. Signature scores are represented as fold-enrichment scores (i.e., mean increase in expression of genes in the signature divided by mean expression of all expressed genes). BRAF expression is log-normalized read counts. Spearman $\rho$ and Bonferroni-corrected q-values are shown for each pairwise comparison

programme reminiscent of normal OPCs. We also identified a MAPK signaling gene programme reflecting Raf/MEK/ERK signaling downstream of the oncogenic KIAA1549-BRAF fusion. Surprisingly, this MAPK signature was not ubiquitously expressed in cancer cells, but instead was confined to a subpopulation of cancer cells that expressed either proliferative or senescence-related genes. Conversely, we identified subpopulations of cancer cells resembling normal mature glia with lower expression of the MAPK signaling programme and of cell cycling programme. Integrating these findings, we propose that the PA cancer cell hierarchy resembles a normal glial maturation process, with cycling progenitor-like cancer cells giving rise to cancer cells that resemble mature glia.

These data indicate that PA cancer cells resemble a developmental spectrum ranging from OPCs to mature glia. While the cells overall resembled OPCs, we also observed

subpopulations of cells expressing a MAPK signaling signature and signatures reminiscent of mature glia. In addition, we observed subpopulations that exhibited intermediate expression of both programme. Intriguingly, F-IHC and RNA ish studies of top marker genes for either gene programme indicate that these gene programme may underlie the long-observed biphasic histopathologic features of PA[42]. These results indicate that the AC-like gene programme is more highly expressed in the piloid, fibrillary component of the tumors, and that expression of the MAPK gene programme is biased towards the loose, microcystic component of the tumors. It stands to reason that one of these populations may give rise to the other, reflecting a developmental process. The fact that the cells expressing the MAPK signaling programme exhibits a higher proportion of cycling cells and expresses progenitor cell-associated transcription factors such as *SOX2* indicates that cells expressing

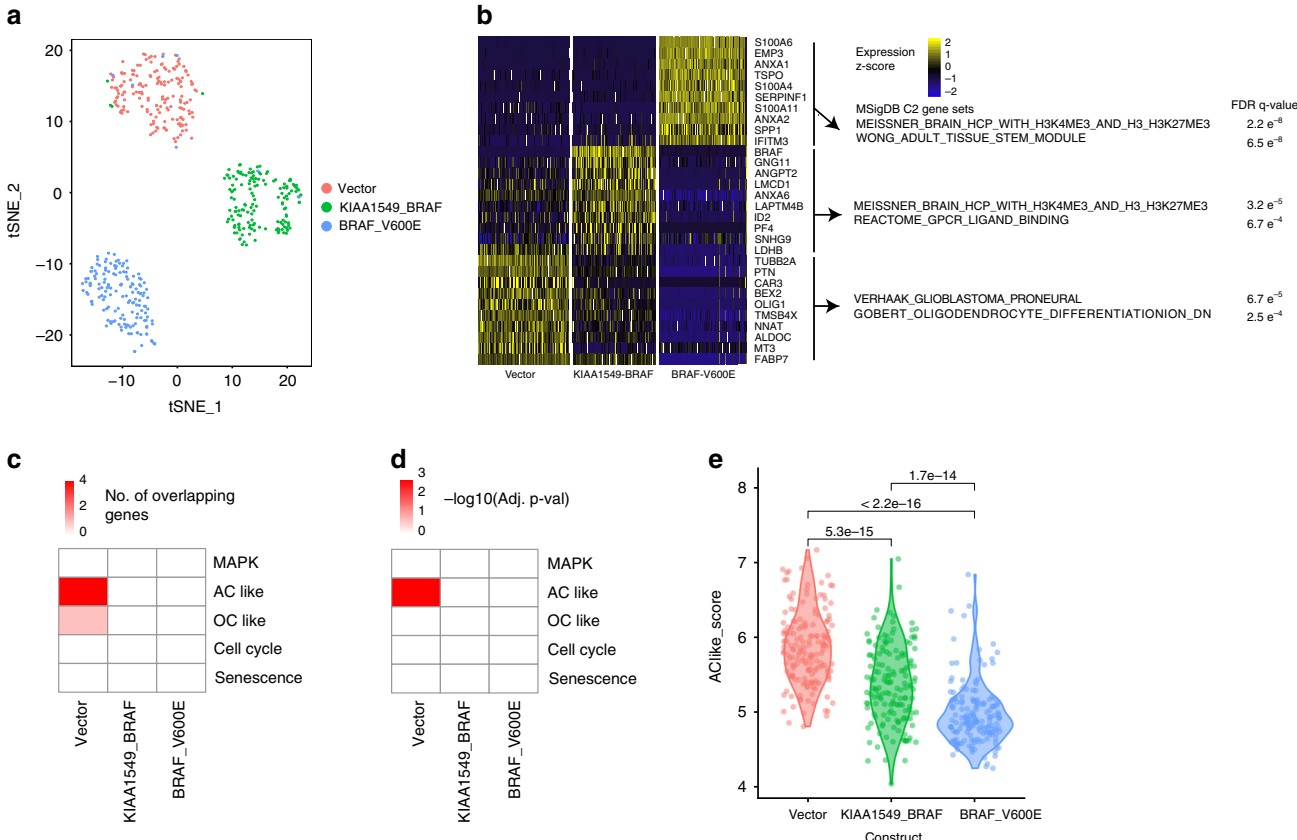

**Fig. 9** Effect of oncogenic BRAF on single mouse neural stem cell transcriptomes. **a** t-SNE plot showing mNSCs expressing KIAA1549-BRAF, BRAF-V600E, or vector control constructs. **b** Heat map showing expression of the top 10 differential genes between mNSCs expressing vector control, KIAA1549-BRAF, or BRAF-V600E constructs. Selected C2 MSigDB gene sets that significantly overlapped with the top 50 differentially expressed genes for each group are shown, along with FDR q-values for significance of overlap using hypergeometric test with Bonferroni correction. **c** Heat map showing number of genes overlapping between list of top 50 genes differentially expressed between mNSC groups and PA-derived gene programme. **d** Heat map showing −log10 of P-value for significance of overlap between gene lists using a Fisher's exact test with a Bonferroni correction. **e** Violin plot showing AC-like gene programme score for mNSCs expressing vector control, KIAA1549-BRAF, or BRAF-V600E constructs. P-values are for Kruskal–Wallis pairwise tests

the MAPK signaling signature may represent an OPC-like progenitor population in PA, which gives rise to an AC-like population. This observation reveals parallels to normal oligodendrocyte differentiation, for which timing of progenitor expansion is tightly regulated by MAPK signaling[43].

Identification of a cellular developmental process in PA raises several therapeutic considerations. We found that a MAPK signaling gene programme was expressed in only a subpopulation of cancer cells, which would suggest that this subpopulation would exhibit differential responses to MEK inhibition compared to the more AC-like cells. Clinically, MEK inhibitors have shown great promise, but complete responses have been rare[17]. The present study raises several testable clinical hypotheses that could explain the heterogeneity of responses to investigational MEK inhibitors and that could guide ongoing clinical investigations. First, MEK inhibition may be inadequate to overcome MAPK signaling in cells with very high levels of MAPK signaling. If this is the case, we predict that tumors with high MAPK gene programme expression may have poor responses to therapy and poor long-term disease control. Second, cells without active MAPK signaling, such as the AC-like+ cells, may be unaffected by MEK inhibition. If so, we predict that tumors with mostly AC-like+ cells would exhibit a poor initial response to MEK inhibition. However, such tumors would exhibit good long-term disease control as the AC-like+ cells are not proliferative. Third, on the

basis of the tumor cell differentiation processes inferred in this study, MAPK+ cells may be able to transition into AC-like+ cells. If so, we predict that tumors that exhibit poor initial responses may exhibit a shift towards higher AC-like+ tumor cell composition between pre-treatment and post-treatment biopsies. If such a process is clinically relevant, it will be critical to determine whether this process is reversible to determine whether further MEK inhibition could be of clinical benefit for these patients. Future correlative and experimental studies informed by the gene programme identified here may provide clarity on these issues and guide the selection of the most efficacious treatment strategies for PA.

Expression of activated BRAF can paradoxically lead to oncogene-induced senescence in vitro, which has been speculated to underlie the relatively indolent biology of BRAF-rearranged PA[34,35]. We found that the highest expression of senescence-related genes was confined to PA cancer cells that highly expressed the MAPK gene programme. Intriguingly, cells highly expressing the MAPK signaling gene programme were also most likely to express a proliferative gene programme, but expression of the senescence and of the proliferative programme occured in mutually exclusive sets of MAPK-activated cells (see Fig. 5e). Future experimental work will be needed to determine whether the dosage of MAPK signaling and/or other cellular factors contribute to proliferative vs. senescent cell fate decisions in this

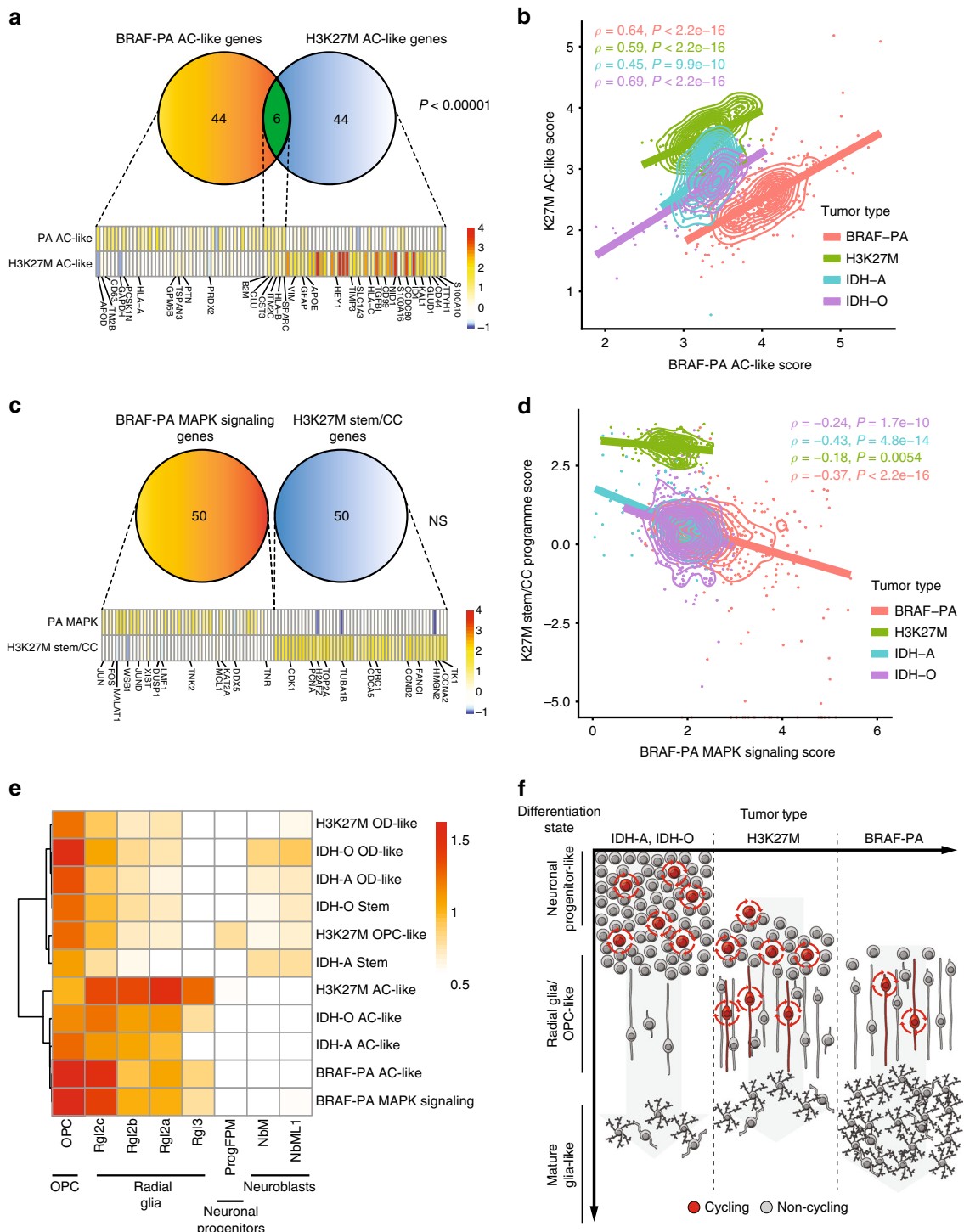

**Fig. 10** Gene programme comparisons between low-grade and high-grade pediatric brain tumors. **a** Venn diagram showing overlap between PA AC-like gene programme and H3K27M AC-like gene programme[4], along with heat map showing average expression of genes from all AC-like programme. P-value represents significance of overlap based on Fisher's exact test. Heat map shows average expression of genes from either gene programme in PA and H3K27M cells. Color scale represents average expression z-score. **b** Single cells from PA tumors, intermediate-grade IDH astrocytomas[2] (IDH-O) and IDH oligodendrogliomas[3] (IDH-A), and from high-grade H3K27M pediatric midline gliomas[4] (H3K27M) are plotted based on expression of the PA AC-like gene programme, and of a H3 AC-like gene programme. Spearman ρ and associated P-values are shown for correlation between the two signatures within cells from each tumor type. **c** Venn diagram showing lack of overlap between PA MAPK signaling gene programme and H3K27M stem/cellcycle gene programme[4]. Heat map shows average expression of genes from either gene programme in PA and H3K27M cells. Color scale represents average expression z-score. **d** Single cells from PA and H3K27M-mutated pediatric midline gliomas are plotted based on expression of the PA MAPK signaling gene programme, and of the stem/cell-cycle gene programme derived from H3K27M tumors. **e** Heat map showing enrichment for human developing midbrain cell type signatures[30] among subpopulations of cells in PA and other glioma types. Color scale shows signature enrichment score. **f** Model for glioma differentiation hierarchies reflecting differences in compartment, abundance of cycling cells, differentiation state, and population structure of PA compared to IDH-O and IDH-A gliomas, and to H3K27M gliomas

context. We speculate that such work could inform therapeutic opportunities to modulate MAPK signaling or other cellular processes to exploit this biology.

PAs exhibit substantial clinical differences from H3K27M-mutant pediatric high-grade gliomas and from IDH-mutant adult intermediate grade gliomas, including a more indolent course with high overall survival and a low incidence of malignant transformation. At the single-cell level, we observed a higher proportion of mature glia-like cells to progenitor-like cells in PAs as compared to these other tumor types. Furthermore, PA cells exhibited a basal gene programme that was reminiscent of normal OPCs, while the higher grade tumors exhibited progenitor populations with relatively more similarities to radial glia and to neuronal stem cells. Both IDH and H3K27M mutations block differentiation by modulating histone methylation marks[44,45], which may explain these differences in developmental hierarchies between the IDH-mutant and H3K27M-mutant tumors and the BRAF-driven PA tumors.

Our findings revealed differences in specific gene programme and in the transcriptional hierarchy between low-grade and high-grade tumors. The data also revealed heterogeneity in expression of mitogenic signaling genes that are currently being investigated as therapeutic targets. A priority for future work will be to determine the effect of such targeted therapies on the PA cellular transcriptional hierarchy.

## Methods

**Tumor acquisition and preparation.** Human tissue analysis complied with all relevant ethical regulations for work with human participants. Informed consent was obtained from all patients and parents at Boston Children's Hospital according to Dana Farber/Harvard Cancer Center Institutional Review Board protocol 10–417. Tissues were collected at the time of surgery and the presence of malignant cells was confirmed by frozen section. Tumor tissues were mechanically and enzymatically dissociated in GentleMACS C tubes using a trypsin or papain-based brain tumor dissociation kit according to the manufacturer's instructions (Miltenyi Biotec).

**KIAA1549-BRAF expressing neural stem cells.** Primary neural stem cell cultures were established from the medial ganglionic eminence of embryonic day 14 murine embryos[38]. Neural stem cells were propagated as neurospheres in DMEM/F12 (50:50 mix) with 1× B27 without vitamin A, and 20 ng per mL EGF[46]. To generate KIAA1549-BRAF expressing cells, mouse neural stem cells were plated on laminin coated dishes and transduced with pBabe (short-isoform) KIAA1549:BRAF[15] (gift of David Jones) or pBabe BRAF-V600E or empty vector control with appropriate antibiotics. BRAF-V600E and KIAA1549:BRAF expressing cells were cultured and propagated in neural stem cell media, except without the presence of growth factors.

**Fluorescence-activated cell sorting.** Dissociated cells were filtered through a 70 μm filter, pelleted 10 min 400 × g, and resuspended in 5 ml serum-free and growth factor-free neurobasal-A media (Life Technologies). Tumor cells were pelleted and then resuspended in 1 ml 1× RBC lysis buffer (ThermoFisher) and incubated for 10 min at room temperature. Cells were washed with 10 mL 1× PBS then resuspended into 100 μL of FACS buffer (1× PBS with 1% BSA), counted, and then diluted to 1–5 million total cells per mL including non-viable. Cells were labeled with 10 μL anti-A2B5-APC (clone: 105HB29, Miltenyi Biotec) and 1.5 μL Calcein Blue AM (Invitrogen, cat# C1429) per 100 μL of cell suspension. Cells were pelleted, washed with 1 mL FACS buffer, and then resuspended into 500 μL FACS buffer containing 0.5 μl of a LIVE/DEAD fixable near IR dead cell stain kit (cat# L-34974). Negative controls with no stain, single, and double stains were used for all tumors. Positive controls for A2B5+ were mouse neural stem cells cultured as neurospheres[38]. Cells were sorted on a SH800S fluorescence activated cell sorter (Sony Biotechnology) or MoFlo Astrios EQ, Cell Sorter (Beckman Coulter). Single cells were sorted by A2B5 status and collected into 96-well plates in approximately equal proportions by A2B5 status. Cells were collected in 5 μl of 1 × Lysis buffer (SmartSeq2 v4 kit, Takara) containing 1% b-mercaptoethanol, spun at 800 × g, snap frozen on dry ice, and stored at −80 °C prior to library preparation.

**Library preparation and sequencing with BRAF spike-in.** cDNA libraries for BT618, BT646, and BT679 were prepared using the a modified SmartSeq protocol[47]. cDNA libraries for BT801, BT827, BT906 we prepared using SmartSeq v4 kit (Takara) with the following modifications. An oligonucleotide specific to a sequence contained in exon 18 of BRAF was used, since this exon is contained in

both long and short isoforms of KIAA1549:BRAF gene products[15]. This BRAF sequence was added 3' to the SmartSeq2 Smart CDS Primer universal adapter sequence. See Supplementary Data 7 for oligonucleotide sequences. This oligonucleotide as added at 13 μM final concentration during 3' primer addition step, at the same time as addition of the Smart CDS Primer IIA. The concentration and size distribution for each cDNA library was evaluated on an Agilent TapeStation 2200 using D5k screentapes. Sequencing libraries from samples with more than 0.3 ng of cDNA greater than 400 bp were prepared using a ¼ Nextera XT reaction[47]. Up to 384 single cell sequencing libraries were pooled and sequenced with paired-end reads using a high output kit on Illumina NextSeq500 or Illumina HiSeq2500 instruments. Sequenced reads were aligned to the UCSC hg19 reference genome assembly using STAR v2.5.1b[48] and gene expression was quantified using RSEM v1.2.28[49] as part of the VIPER snakemake pipeline[50]. Transcripts reflecting discordant mapping of reads between experiments were manually removed for downstream analyses. KIAA1549-BRAF gene fusions were identified in the RNA-seq data using STAR-Fusion v0.5.4[51]. The BRAF spike-in workflow was tested for systematic differences introduced into the data by examining mouse neural stem cells processed with and without the BRAF oligonucleotide spike-in (n = 6 for each). The resulting transcriptomic data from these cells clustered together based on principal component analysis and hierarchical clustering (Supplementary Fig. 13), indicating that data is comparable between protocols.

**Detection of BRAF fusion from single cell cDNA libraries.** qPCR was performed on remaining cDNA libraries after sequencing if any material was available following NGS. Clinical detection of KIAA1549-BRAF fusion breakpoints were determined by OncoPanel targeted sequencing. Taqman qPCR probes targeting the most common BRAF fusion breakpoints 15–9, 16–9, or 16–11 were used[49], see Supplementary Data 7 for sequences. cDNA from BT827 and BT618 had been exhausted for RNA-seq analysis and was not available to perform qPCR. BT646 contained a complex rearrangement involving multiple breakpoints that linked KIAA1549 to BRAF that was not amenable to this qPCR approach.

**Data quality control, dimension reduction, and clustering.** All processed data was analyzed in R v3.4.3. Data were initialized using functions in the Seurat package[52] with default parameters, unless specified otherwise, as follows. Transcripts per million data normalized for each cell by the total expression, then multiplied by a scale factor of 10,000, and then log transformed using the LogNormalize function. Data were scaled using the ScaleData function. To remove sources of variation that did not reflect common features of PA tumors, linear regression was performed on number of detected genes and on tumor of origin. For quality control filtering, cells with <1000 expressed genes or >4% mitochondrial genes were excluded from analysis and genes expressed in <10 cells were excluded from analysis. Variable genes with 0.25 to 5 log-normalized reads and >1 standard deviation were subjected to PCA in Seurat. The first 7 principal components were used for t-SNE generation for all cells. Shared nearest neighbor clustering was performed using the FindClusters function with k parameters = 150 and using the first 7 principal components. The first 5 principal components were used for t-SNE generation for the cancer cell subset. NMF was performed using R package NNMF[53]. Rare A2B5+ cells that clustered with the immune cells were inspected for expression of PA cancer cell differential genes and for KIAA1549-BRAF expression, and fourteen discordant cells without expression of these markers were removed from the A2B5+, BRAF+ group. Rare A2B5+ cells expressing immune genes were removed from cancer cell analyses by filtering out cells expressing >1.5 log-normalized GZMB reads. Gene set enrichment analysis was performed using the javaGSEA desktop application[54].

**Data visualization.** Bar plots, PCA plots, volcano plots, lineage score plots, and violin plots were generated using ggplot2[55]. Heat maps were generated using the pheatmap function on the scaled data matrix. T-SNE plots were generated using the FeaturePlot and TSNEPlot functions in Seurat with default settings.

**Inferral of CNVs from single cell RNA-seq data.** Large-scale chromosomal CNVs were inferred from expression data as using the inferCNV package[1]. Expression for each gene was averaged over 100-gene windows. Averaged expression intensity was then compared to all other cancer cells as a reference.

**Estimation of contribution of cell types to bulk expression.** To estimate contribution of different cell subtypes to bulk sequencing profiles, differential gene lists were first derived for all cancer cell compartments found in the single cell RNA-seq data (tumor, microglia, T cell, and macrophage) using Wilcoxon rank sum t-tests. Genes with FDR < 0.05 (Bonferroni correction) were used for each differential gene list. For each bulk expression profile, enrichment scores for each compartment were calculated by dividing the average z-scored expression of all genes in that compartment's differential gene list, divided by the average expression of all genes. The estimated percent contribution of each compartment was calculated by dividing the enrichment score for each compartment into the sum of enrichment scores for all compartments. Genes that did not overlap between the scRNA-seq platform and the bulk expression analysis platform were discarded for this analysis.

**Enrichment score calculation and visualizations**. To calculate enrichment scores for normal brain gene signatures in single cells, the mean scaled expression score for all genes in the signature was divided by the mean scaled expression score for all genes. Mean enrichment scores for clusters and for different types of tumors were calculated in the same fashion, except the mean expression for each gene across all cells in the cluster was used instead of the expression of each gene in a single cell. Gene signatures for microglia and macrophage subtypes from a single cell RNA-seq atlas of mouse microglia[25] were the list of upregulated genes for a given subtype (>1.5 fold change and FDR $q$-value < 0.05 using Wilcoxon rank sum test and Bonferroni correction). Cluster 7a could not be analyzed in this way since only one gene was upregulated in that gene list, and that gene did not have a human ortholog that was expressed in our scRNA-seq dataset. Each microglia/macrophage signature enrichment scores were normalized to the mean enrichment score for the same signature in the tumor clusters. BioMart was used to convert mouse gene names to human ortholog gene names for comparisons to mouse gene lists[56]. Gene programme were derived from the top or bottom 50 genes in principal component variables. Tumor subpopulations were defined as cancer cells with enrichment scores of >2.5 for a given gene programme. Cell cycle and senescence scores were determined by testing for enrichment of cell cycle[37] and for senescence[36] gene sets from the C2 curated gene set in MSigDB (http://software.broadinstitute.org/gsea/msigdb/index.jsp). For heat maps comparing normal brain gene signature enrichment between different tumor types, 500 cancer cells of each type were randomly sampled and the mean enrichment scores for those cells was calculated.

**RNA in situ staining**. RNA in situ staining was performed on freshly cut unstained formalin-fixed, paraffin embedded slides[4] using probes RNAscope Hs-APOD (cat no. 445171) and RNAscope Hs-JUN (cat no. 470541) (ACD Biotech).

**Fluorescence immunohistochemistry**. A multiplexed F-IHC was performed[57]. The antibody panel consisted of Olig2 (R&D System, Goat polyclonal, AF2418) with CY5.5, GFAP (Cell Signaling Techology, Mouse monoclonal, Clone GA5, #3670) with FITC, and Ki-67 (Dako, Mouse monoclonal, Clone MIB-1, M7240) with CY3. Whole slide Images were acquired from stained slides using a Perkin Elmer Vectra 3 imaging system (PerkinElmer, Inc.), and InForm software (version 3.4.3, PerkinElmer) was used to unmix the signals. Twenty high power fields (20×) were analyzed utilizing Halo Image Analysis platform (Indica Labs) for each tumor. The thresholds for the markers were set, respectively, based on the staining intensity, by cross reviewing 20 images. Cells with the intensity above the setting threshold were defined as positive.

**Bulk tumor profiling**. BRAF fusion calls, CNVs, and mutation calls from bulk tissues were extracted from OncoPanel[58] data. DNA was isolated from tissue containing at least 20% tumor nuclei. Agilent SureSelect hybrid capture kit was used to target exonic DNA sequences of 300 cancer genes and 113 introns across 35 genes for rearrangement detection prior to Illumina HiSeq 2500 sequencing.

**Cell trajectory construction**. Cell trajectory visualization and inference of pseudotime values for cancer cells were performed using Monocle v.2.6.4[33] in R. Cancer cells that had been filtered as described above were input into cell trajectory analyses. The top 1000 differentially expressed genes as determined by the differentialGeneTest function were used for cell ordering. The cell trajectory was learned using the reduce Dimension function using the DDRTree dimension reduction method, the first five principal components, and tumor of origin and number of expressed genes as residuals. The plot_cell_trajectory function was used to plot the cell trajectory, with color_by and markers calls used to specify the parameter to account for coloring of the cells.

**Mouse neural stem cell scRNA-seq**. mNSCs expressing vector control, KIAA1549-BRAF, or BRAF-V600E were generated using pBabeNeo retroviral vectors and cultured in growth factor-containing media[38]. Cells were trypsinized and subjected to scRNA-seq as described for human cells above, with the following modifications. BRAF oligonucleotide spike-in and qPCR was not performed for mouse scRNA-seq. Transcriptomes were aligned to UCSC mouse mm10 assembly. Data were generated for 576 mNSCs before quality filtering. Cells with <3000 genes or >3% mitochondrial genes were removed for data filtering. Genes expressed in <10 cells were removed for data filtering. For comparisons between human and mouse systems, gene lists of interest were converted from human to mouse gene names as described above. Lists of differentially expressed genes were compared to C2 curated gene sets using the Compute Overlaps function in MSigDB.

**Statistics**. FDR-adjusted $q$-values were determined for differential expression analyses using the Benjamini–Hochberg method. Student's $t$-test was used to test differences in mean expression of markers in flow cytometry analyses. Wilcoxon rank sum tests were used to test for differences in mean gene signature enrichment scores between groups, and to calculate $P$-values for genes differentially expressed between clusters or between tumor and tumor-associated cells. For gene programme overlap analysis, Fisher's exact test was used to test significance of gene programme overlaps assuming a genome size of 18,000 genes. Spearman's rho was

calculated to assess correlations between gene programme scores among cell populations. To correct for testing multiple hypotheses when examining pairwise correlations between different gene programme, or for computation of the significance of overlaps between multiple pairs of gene lists, Bonferroni-corrected q-values were reported.

**Reporting summary**. Further information on research design is available in the Nature Research Reporting Summary linked to this article.

## Data availability

The human RNA sequencing data has been deposited in the dbGaP repository under the accession number phs001854.v1.p1. Expression data and metadata for tumor of origin, cluster, and t-SNE coordinates for single cells have been uploaded to the Broad Institute Single Cell Portal under accession number SCP271. The mouse neural stem cell RNA sequencing data, expression matrix, and metadata has been deposited to the Single Cell Portal under the accession number SCP468. The source data underlying Fig. 6a–j are provided as a Source Data file. The datasets for H3K27M mutant pediatric midline gliomas[4], oligodendrogliomas[2], and intermediate grade astrocytomas[3] referenced in the study are available from the Single Cell Portal with accession numbers SCP147, SCP12, and SCP50, respectively. Normal adult cortex gene signatures were derived from marker gene lists found in Fig. 1 of a scRNA-seq atlas of normal adult cortex[31]. Developing midbrain gene signatures were derived from the marker gene matrix found in Table S2 of a publication by La Manno and colleagues[30]. Each midbrain cell type gene signature comprised the list of all genes expressed in that cell type in the marker gene matrix. Developing cortex gene signatures were derived from the differential gene list found in Table S5 of a publication by Nowakowski and colleagues[32]. For the developing cortex gene sets, only the differential genes with $P < 0.0005$ (non-parametric Wilcoxon rank sum test) were used for the gene signature for each cell type in order to restrict gene signatures to ~200 genes. All the other data supporting the findings of this paper are available within the article, the supplementary information files and from the corresponding author upon reasonable request.

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

## Acknowledgements

We would like to thank and acknowledge the Dana-Farber PLGA programme, Boston Children's Hospital Division of Neuro-pathology, staff of the Dana-Farber Cancer Institute Flow Cytometry Core, and members of the Bandopadhayay and Beroukhim Laboratories for their helpful discussions. We would like to thank Eric Smith for assistance with illustrations. We acknowledge the following funding sources: The Pediatric Low-Grade Astrocytoma Fund at the Pediatric Brain Tumor Foundation (R.S., L.G., D.H.K., M.W.K., K.L.L., C.S., P.B., R.B.), Stop and Shop Pediatric Brain Tumor Program (P.B., M.W.K.), a Conquer Cancer Foundation and Strike 3 Foundation ASCO Young Investigator Award (Z.J.R.), a Michael Mosier Defeat DIPG and ChadTough Foundation Fellowship (Z.J.R.), NIH PO1CA142536 (R.S.,. C.S., K.L.L., R.B.), K99CA201592 (P.B.), R01CA188228 (K.L.L. and R.B.), R01CA142536 (K.L.L. and R.B.), and R01CA219943 (K.L.L. and R.B.), F32CA180653 (B.R.P.), St Baldrick's Foundation (P.B.), Friends of DFCI (Z.J.R., B.P., G.B., P.B.), Team Jack Foundation (P.B., M.W.K., R.B., L.G.), Andrysiak Fund for LGG (M.W.K.), Jared Branfman Sunflowers For Life Fund For Pediatric Brain And Spinal Cancer Research (P.B. and R.B.), The Gray Matters Brain Cancer Foundation (R.B.), Ian's Friends Foundation (R.B.), Sontag Foundation (K.L.L., R.B.), the Searle Scholars Program (A.K.S.), the Beckman Young Investigator Program (A.K.S.), the Pew-Stewart Scholars (A.K.S.), a Sloan Fellowship in Chemistry (A.K.S.), NIH grants 2U19AI089992 (A.K.S.), 1U54CA217377 (A.K.S.), 2P01AI039671 (A.K.S.), 5U24AI118672 (A.K.S.), 2RM1HG006193 (A.K.S.), 1R33CA202820 (A.K.S.), 2R01HL095791 (A.K.S.), 1R01AI138546 (A.K.S.), 1R01HL134539 (A.K.S.), 1R01DA046277 (A.K.S.), and Bill and Melinda Gates Foundation grants OPP1139972 (A.K.S.), OPP1137006 (A.K.S.), OPP1116944 (A.K.S.).

## Author contributions

P.B. and R.B. contributed equally to this work. B.P., G.B., A.K.S., P.B. and R.B. conceived the study. Z.J.R., B.P., G.B., A.K.S., P.B. and R.B. designed the experiments and interpreted results. Z.J.R., B.P., A.L.C., K.Q. and G.B. collected single cells and generated single-cell sequencing data. Z.J.R. and P.K. performed computational analyses. B.P. performed Spike-in-Seq qPCR. B.P., G.B., J.W. and Y.S. performed in vitro experiments. Z.T.H., L.G., J.D. and Y.H. provided sequencing, flow cytometry, F-IHC and RNA ish support. S.B., R.J., K.P. and K.L.L. provided tissue collection and processing support. C.S., H.M., J.A., O.R.R., R.S., D.H.K., M.F., M.S., A.R., C.S., M.K. and L.G. provided experimental support, analytical support, and critical feedback. Z.J.R., B.P., P.B. and R.B. wrote the paper with feedback from other authors.

## Additional information

**Competing interests:** P.B. and R.B. received grant funding from Novartis and R.B. has

received consulting fees from Novartis. R.B. consults for and owns shares in Ampressa. A.R. is a scientific advisory board member for ThermoFisher Scientific, Syros Pharmaceuticals and Driver Group. A.K.S have received compensation for consulting and SAB membership from Honeycomb Biotechnologies, Cellarity, Cogen Therapeutics, and Dahlia Biosciences. M.W.K. is now an employee of Bristol-Myers Squibb. G.B. is now an employee of Roche. All the remaining authors declare no competing interests.

