## [Peer Review File · Nature Communications]

Reviewers' comments:

Reviewer #1 (Remarks to the Author):

Reitman et al. performed single-cell RNA-seq on 6 pilocytic astrocytomas. They then clustered those data, identified tumor-infiltrating immune cells, performed several differential expression tests among subpopulations and higher grade pediatric gliomas, as well as principle components analysis and other varied bioinformatics.

The conclusions drawn are modest and supported by the data. The tissue used for profiling are rare specimens. The comparisons to other gliomas at the single-cell level are useful. The data will be useful to researchers working in brain tumors and other cancers. The manuscript is entirely descriptive but from that perspective is fairly complete. I would ask the authors to consider the following:

1. Can the authors use RNA velocity to infer lineage relationships between their identified putative stem-like population and other cell types?
2. Consider validating co-expression of stem-cell, proliferation markers using RNA-FISH and/or IF; likewise for more differentiated cell types.
3. What contributes to the genomic instability in the specimens exhibiting large scale CNVs?
4. What is the impact of this study on PA therapy? The comparisons to other diseases are interesting, but does this study have any impact beyond basic science?

Reviewer #2 (Remarks to the Author):

The manuscript by Reitman and colleagues is the first one to report on scRNA-Seq data from pilocytic astrocytoma, the overall most common brain tumor in children. The investigators subjected 6 PAs from different brain regions to scRNA-Seq and compared with data that they had previously obtained and published on higher-grade gliomas in children and adults. One of the most interesting results is that the differences in transcriptomes between supratentorial and infratentorial PAs can largely be explained by differential proportions of immune cells. The other important finding is the gradient of differentiation within PA cells.

While the comparisons between PA and high-grade gliomas are interesting in part, they also didn't reveal something really surprising/novel. The probably most interesting question for PA, namely the intratumoral heterogeneity of senescent versus non-senescent cells (in conjunction with MAPK activation) was unfortunately not (fully) addressed. Actually there is not a single word about MAPK-induced senescence in the paper. Additionally: How does this correlate with BRAF fusion transcript abundance?

The interpretation of the heterogeneous response to MEK inhibition is not entirely clear since the authors at the same time claim that MAPK activity is associated with proliferation, so these cells should be targeted? Do the authors imply plasticity such that MAPK high cells can escape MEK inhibition by morphing into MAPK low cells? Was any of this functionally analyzed in the short term culture systems the authors have pioneered?

Minor comments:

@introduction: some of the working about clinical associations of PAs is a bit misleading, e.g., "adult PA patients... rarely succumb to their disease". This is equally true for children. The overall progression rate is also certainly not 80% (which might be the interpretation when people read the text as it currently stands).

@amplification: The authors seem to describe single copy-number gains here rather than amplifications?

@Figure 2D: the annotation for the identified clusters is missing in the figure + figure legend.

@BT827: Was this the tumor with a non-canonical BRAF duplication? What is the fusion partner here and why can it not be seen in the scRNA-Seq data?

Stefan Pfister

Replies to referee comments

NCOMMS-19-01083-T

Mitogenic and progenitor programs in single pilocytic astrocytoma cells

Reviewer #1 (Remarks to the Author):

Reitman et al. performed single-cell RNA-seq on 6 pilocytic astrocytomas. They then clustered those data, identified tumor-infiltrating immune cells, performed several differential expression tests among subpopulations and higher grade pediatric gliomas, as well as principle components analysis and other varied bioinformatics.

The conclusions drawn are modest and supported by the data. The tissue used for profiling are rare specimens. The comparisons to other gliomas at the single-cell level are useful. The data will be useful to researchers working in brain tumors and other cancers. The manuscript is entirely descriptive but from that perspective is fairly complete.

We thank the Reviewer for their positive comments.

I would ask the authors to consider the following:

1. Can the authors use RNA velocity to infer lineage relationships between their identified putative stem-like population and other cell types?

We thank the Reviewer for this suggestion. We first attempted to analyze our data using the RNA velocity software package. Unfortunately, we encountered technical difficulties that prevented us from successfully applying this algorithm to our RNA-seq data, despite multiple attempts and communication with the primary authors of the report¹ describing the RNA velocity algorithm.

However, we agree with the Reviewer that inferral of lineage relationships is important. Although RNA velocity addresses changes that occur over hours, lineage relationships are likely to occur over longer periods of time. Such temporal changes are likely to be better addressed by Pseudotime analysis. Pseudotime analysis models cells as transitioning between a starting state and one or more end states, with many cells distributed in a trajectory between them.

Applying this approach to the PA scRNAseq data shows that cells predominantly expressing the MAPK gene program branch into cells expressing either of the mature glia gene programs. These analyses were consistent with our previous findings. We now include the Pseudotime analyses in Figure 7 and the Results section:

To examine whether a developmental process is plausible based on our scRNA-seq data, we used a reverse graph embedding approach² to construct a cellular trajectory from the cancer cell RNA-seq profiles. This analysis ordered cells within the trajectory in

“pseudotime”, ie cells with low pseudotime values occupy early states in the inferred developmental process and cells with high pseudotime values occupy later states (Fig. 7a). Cells from all six tumors were well-represented throughout different regions of the trajectory (Fig. 7b). The initial low-pseudotime cells within this trajectory had high MAPK signaling gene program scores and highly expressed MAPK genes (Fig. 7c, Supplementary Fig. 11a). In contrast, high-pseudotime cells later in the trajectory had increasing expression of AC-like gene program scores and of AC-like genes (Fig. 7d, Supplementary Fig. 11b). Most cells did not score highly for the OC-like gene program, but the few that did were mostly in a distal branch with high pseudotime values (Supplementary Fig. 11c,d). These results support a developmental process in PA cancer cells.’

We also attempted to use our neural stem cell model systems to explore the causal effects of driver oncogene expression on cell lineage gene programs. These experiments involved single cell RNA-seq analysis of almost 500 mouse neural stem cells (mNSCs) which we transduced to express KIAA1549-BRAF, BRAFv600e or vector controls. These results showed that expression of oncogenic BRAF decreases expression of the astrocyte-like gene program, but not other gene programs. These data complement the above analysis by showing a causal role for BRAF fusion that may underlie the lineage relationships observed in tumors. We refer to these data in Figure 9 and the Results section:

‘Since we observed differential expression of oncogenic BRAF in cancer cells with different cell states, we hypothesized that expression of oncogenic BRAF may influence the state of PA cancer cells by modulating one or more of the gene programs identified above. To test this hypothesis, we examined single cell transcriptomes of mouse neural stem cells (mNSCs) that we engineered to express the KIAA1549-BRAF fusion³. In parallel, we examined vector control mNSCs and mNSCs engineered to express BRAF-V600E, another common BRAF alteration found in pediatric gliomas³. Single cell RNA-seq data were generated for 487 mNSCs after quality filtering (n=170 vector control, n=154 expressing KIAA1549-BRAF, and n=163 expressing BRAF-V600E). mNSCs expressing each construct clustered separately from each other in transcriptional space (Fig. 9a). The top differentially-expressed genes for mNSCs expressing each construct were identified (Fig. 9b). Comparison of differentially expressed gene lists to the C2 MSigDB database revealed that genes upregulated in mNSCs expressing oncogenic BRAF constructs overlapped with genes in high-CpG-density promoters bearing histone H3 dimethylation at K4 and trimethylation at K27 (H3K27me3) in brain⁴. In contrast, genes differentially expressed in the vector control cells significantly overlapped with a proneural glioma signature⁵. To determine whether any of the constructs may upregulate or downregulate gene programs identified in our human PA single cell RNA-seq analysis, we computed overlaps between the list of genes differentially upregulated in mNSCs expressing each construct and the PA-derived gene programs (Fig. 9c). Only the AC-like program and the genes upregulated in the vector control cells (and thus downregulated in the KIAA1549-BRAF and BRAF-V600E cells) demonstrated a significant overlap (Fig. 9d, FDR q-val 0.002). Indeed, AC-like gene program scores were significantly lower for mNSCs expressing the BRAF constructs compared to controls (Fig. 9e). This analysis demonstrates that oncogenic BRAF expression can

oppose expression of mature glia gene programs, indicating that dynamics of BRAF expression may contribute to heterogeneity in cancer cell states found in PAs. Of note, we could not detect a further increase in MAPK signaling gene program expression in the BRAF-expressing mNSCs compared to controls. We speculate that this may be due to the necessity to culture mNSCs in growth factors that exogenously activate MAPK signaling in these growth factor-dependent stem cells³.

2. Consider validating co-expression of stem-cell, proliferation markers using RNA-FISH and/or IF; likewise for more differentiated cell types.

We have now validated co-expression of a progenitor/stem-cell marker (OLIG2), a differentiated cell marker (GFAP), and a proliferative marker (Ki-67) using fluorescence immunohistochemistry (F-IHC). This analysis confirmed expression of top markers of different gene programs identified in our scRNA-seq analysis and also linked the AC-like gene program, marked by the cell surface marker GFAP, to PA tumor components that exhibit a fibrillary morphology. In contrast, the marker OLIG2 is linked to microcystic components of the tumors.

These intriguing results indicate that the gene programs identified by scRNA-seq underlie the long-observed biphasic histology of pilocytic astrocytomas⁶.

We include these data in Figure 6 and the Results section. Furthermore, we now also include further Discussion of the association of different lineages with the biphasic histology of PAs:

‘Intriguingly, F-IHC and RNA ish studies of top marker genes for either gene program indicate that these gene programs may underlie the long-observed biphasic histopathologic features of PA⁶. These results indicate that the AC-like gene program is more highly expressed in the piloid, fibrillary component of the tumors, and that expression of the MAPK gene program is biased towards the loose, microcystic component of the tumors.’

These results also validate the findings of our scRNA-seq analysis at the protein level. These protein-level data complement the RNA *in situ* hybridization data included in the initial submission that showed transcript-level expression of top markers of the PA gene programs, *JUN* and *APOD*.

3. What contributes to the genomic instability in the specimens exhibiting large scale CNVs?

We thank the Reviewer for this intriguing question. This led us to test whether clinical features of the patients, mutational burden, or mutational status of genes that could be associated with genomic instability were associated with the presence or absence of copy number variants in our cohort (see Table below). However, we could not identify a clinical characteristic or gene mutation that was recurrently or significantly associated

with the presence or absence of copy number alterations in our cohort. These data are included in Supplementary Tables 1-2 and Supplementary Fig. 2.

Tumor	BT618	BT646	BT679	BT801	BT827	BT906
Copy number variants inferred from scRNAseq data	Chr 12 gain Chr 19 loss	Chr 5, 7 gain	none	none	Chr 11q gain	Chr10 gain
Nucleotide variants potentially associated with genomic instability	CDK4 c.752C>A (p.P251H)			BRCA2 c.1059_1060insT (p.V355Cfs*3); BRCA2 c.10082A>C (p.Q3361P)		KDM6B c.2253_2255delCAC (p.T754del); SETD2 c.1877C>G (p.T626S)
Total exonic mutations	0	5	0	15	0	5
Age at diagnosis	4	14	8	4	6	14
Patient sex	M	M	F	F	M	F

4. What is the impact of this study on PA therapy? The comparisons to other diseases are interesting, but does this study have any impact beyond basic science?

We agree that the immediate impact of this work primarily relates to understanding PA biology. However, our findings do frame questions for future studies that may explain observations made in the clinic and that may have direct clinical impact. We have now included these concepts in a revised section of the Discussion:

‘Identification of a cellular developmental hierarchy in PA raises several therapeutic considerations. We found that a MAPK signaling gene signature was expressed in only a subpopulation of cancer cells, which would suggest that this subpopulation would exhibit differential responses to MEK inhibition compared to the more AC-like cells. Clinically, MEK inhibitors have shown great promise, but complete responses have been rare⁷. The present study raises several testable clinical hypotheses that could potentially explain the heterogeneity of responses to investigational MEK inhibitors. First, MEK inhibition may be inadequate to overcome MAPK signaling in cells with very high levels of MAPK signaling. If this is the case, we predict that tumors with high MAPK gene program expression may have poor responses to therapy and poor long-term disease control. Second, cells without active MAPK signaling, such as the AC-like+ cells, may be unaffected by MEK inhibition. If so, we predict that tumors with mostly AC-like+ cells would exhibit a poor initial response to MEK inhibition. However, such tumors would exhibit good long-term disease control as the AC-like+ cells are not proliferative. Third, on the basis of the tumor cell differentiation processes inferred in this study, MAPK+ cells may be able to morph into AC-like+ cells. If so, we predict that tumors that exhibit poor initial responses may exhibit a shift towards higher AC-like+ tumor cell composition between pre- and post-treatment biopsies. If such a process is clinically relevant, it will be critical to determine whether this process is reversible/plastic to determine whether further MEK inhibition would be of clinical benefit for these patients. Future correlative and experimental studies informed by the gene programs identified here may provide clarity on this issue and guide the selection of the most efficacious new treatments for PA.’

Reviewer #2 (Remarks to the Author):

The manuscript by Reitman and colleagues is the first one to report on scRNA-Seq data from pilocytic astrocytoma, the overall most common brain tumor in children. The investigators subjected 6 PAs from different brain regions to scRNA-Seq and compared with data that they had previously obtained and published on higher-grade gliomas in children and adults. One of the most interesting results is that the differences in transcriptomes between supratentorial and infratentorial PAs can largely be explained by differential proportions of immune cells. The other important finding is the gradient of differentiation within PA cells.

We thank Dr. Pfister for kindly noting the important findings regarding differences in transcriptomes tumors in different brain locations and the differentiation gradient in PA cells.

While the comparisons between PA and high-grade gliomas are interesting in part, they also didn't reveal something really surprising/novel. The probably most interesting question for PA, namely the intratumoral heterogeneity of senescent versus non-senescent cells (in conjunction with MAPK activation) was unfortunately not (fully) addressed. Actually there is not a single word about MAPK-induced senescence in the paper.

We agree with Dr. Pfister that the presence of oncogene induced senescence is important to the biology of pediatric low-grade gliomas and we thank him for his suggestion. To comprehensively map the proliferative and senescent cells within the transcriptomic population structure of PAs, we have now examined co-expression of senescence, proliferation, MAPK, and mature glia gene signatures. We find that proliferative and senescent cells represent mutually exclusive and relatively small subpopulations within the MAPK+ compartment. We include these data in Figure 5e and Figure 8 and in the Results:

‘It has previously been shown that expression of activated BRAF can paradoxically lead to oncogene-induced senescence *in vitro*, which may explain the relatively favorable clinical outcomes associated with PA^{8,9}. We therefore sought to determine which PA cells were proliferative and which were senescent within the transcriptional population structure that we identified within PA. To do so, we examined whether senescence¹⁰ or cell cycling¹¹ gene programs were correlated with either the MAPK signaling or mature glia gene programs among cancer cells in our dataset (Fig. 8a,b). We found that the MAPK and senescence programs were the most strongly correlated pair in this analysis (Spearman $\rho = 0.24$, Bonferroni-adjusted $q < 10^{-6}$). The MAPK gene program score was also significantly correlated with BRAF expression ($\rho = 0.14$, $q = 0.008$) and with the cell cycling score ($\rho = 0.12$, $q = 0.04$). Cells expressing high levels of MAPK signaling and low AC- or OC-like scores were enriched for cells expressing markers of the cell cycle (Fig. 5e). As expected, the senescence and cell cycling scores were not correlated ($\rho = 0.015$, NS). These results demonstrate that proliferating and senescent cancer cells are mutually exclusive subpopulations within a compartment of PA cancer cells that

expresses a MAPK gene program. Together, these findings link a MAPK transcriptomic program with senescence *in vivo*. The findings also identify transcriptomic features of a proliferative tumor cell compartment that should be the focus of correlative studies of investigational therapies.'

We consider these findings in light of this important topic in the Discussion:

'Expression of activated BRAF can paradoxically lead to oncogene-induced senescence *in vitro*, which has been speculated to underlie the relatively indolent biology of BRAF-rearranged PA^{8,9}. We found that the highest expression of senescence-related genes was confined to PA cancer cells that highly expressed the MAPK gene program. Intriguingly, cells highly expressing the MAPK signaling gene program were also most likely to express a proliferative gene program, but expression of the senescence and of the proliferative programs appeared to occur in mutually exclusive sets of MAPK-activated cells (see Fig. 5e). Future experimental work will be needed to determine whether the dosage of MAPK signaling and/or other cellular factors contribute to proliferative vs. senescent cell fate decisions in this context. We speculate that such work could inform therapeutic opportunities to modulate MAPK signaling or other cellular processes to exploit this biology.'

Additionally: How does this correlate with BRAF fusion transcript abundance?

We now show that BRAF fusion is significantly more highly expressed in the MAPK+ cells (new Supplementary Fig. 10). Further, BRAF expression correlates with MAPK gene program expression score (see the BRAF and MAPK subpanel within new Fig. 8b).

We did not detect significant correlations between BRAF expression and signatures of either proliferation or of senescence (see BRAF-senescence and BRAF-cell cycle subpanels of the new Fig. 8b).

The interpretation of the heterogeneous response to MEK inhibition is not entirely clear since the authors at the same time claim that MAPK activity is associated with proliferation, so these cells should be targeted? Do the authors imply plasticity such that MAPK high cells can escape MEK inhibition by morphing into MAPK low cells?

We apologize that this was not clear. We believe that the present study raises several testable clinical hypotheses, one or more of which could explain heterogeneous responses to MAPK-directed therapies seen in the clinic. Testing these predictions has the potential to improve the management of PA in the clinic. We note that considerable correlative and experimental work will be needed to test these hypotheses. We have now clarified these concepts in a revised section of the Discussion:

'Identification of a cellular developmental process in PA raises several therapeutic considerations. We found that a MAPK signaling gene program was expressed in only a subpopulation of cancer cells, which would suggest that this subpopulation would

exhibit differential responses to MEK inhibition compared to the more AC-like cells. Clinically, MEK inhibitors have shown great promise, but complete responses have been rare⁷. The present study raises several testable clinical hypotheses that could explain the heterogeneity of responses to investigational MEK inhibitors and that could guide ongoing clinical investigations. First, MEK inhibition may be inadequate to overcome MAPK signaling in cells with very high levels of MAPK signaling. If this is the case, we predict that tumors with high MAPK gene program expression may have poor responses to therapy and poor long-term disease control. Second, cells without active MAPK signaling, such as the AC-like+ cells, may be unaffected by MEK inhibition. If so, we predict that tumors with mostly AC-like+ cells would exhibit a poor initial response to MEK inhibition. However, such tumors would exhibit good long-term disease control as the AC-like+ cells are not proliferative. Third, on the basis of the tumor cell differentiation processes inferred in this study, MAPK+ cells may be able to morph into AC-like+ cells. If so, we predict that tumors that exhibit poor initial responses may exhibit a shift towards higher AC-like+ tumor cell composition between pre- and post-treatment biopsies. If such a process is clinically relevant, it will be critical to determine whether this process is reversible to determine whether further MEK inhibition could be of clinical benefit for these patients. Future correlative and experimental studies informed by the gene programs identified here may provide clarity on these issues and guide the selection of the most efficacious new treatment strategies for PA.'

Was any of this functionally analyzed in the short term culture systems the authors have pioneered?

We thank the reviewer for bringing up these points. As suggested, we have now examined this issue using our mouse neuronal stem cell culture system³. We show that expression of oncogenic BRAF found in pilocytic astrocytomas decreases expression of the astrocyte-like gene program, but not other gene programs (new Figure 9). We thank the Reviewer for suggesting these experiments.

We note that we have attempted to perform scRNA-seq to explore the effects of clinically relevant treatments on short-term primary tumor culture systems. However, our initial experiments showed low scRNA-seq data quality in these experiments. We hypothesize that this low data quality is due to prolonged *ex vivo* culture during drug treatment. Our goal is to further optimize single cell genomic analyses on primary culture systems in future studies.

Minor comments:

@introduction: some of the working about clinical associations of PAs is a bit misleading, e.g., "adult PA patients... rarely succumb to their disease". This is equally true for children.

We have reworded these sections as follows: "PAs that are incompletely resected tend to recur during childhood, but childhood PA patients usually do not succumb to their disease^{12,13}"

The overall progression rate is also certainly not 80% (which might be the interpretation when people read the text as it currently stands).

We agree that this was misleading, and we have removed this sentence.

@amplification: The authors seem to describe single copy-number gains here rather than amplifications?

The reviewer is correct. We have clarified the text to refer to these events as copy-number gains rather than amplifications.

@Figure 2D: the annotation for the identified clusters is missing in the figure + figure legend.

We have added the annotation for the clusters to the figure and legend.

@BT827: Was this the tumor with a non-canonical BRAF duplication? What is the fusion partner here and why can it not be seen in the scRNA-Seq data?

We apologize for not making this clear. BT827 had a classical BRAF duplication (BT646 had the non-canonical BRAF duplication). However, in the case of BT827, single cell cDNA library material had been exhausted by RNA-seq and was not available to perform the qPCR component of the BRAF Spike-in-Seq workflow. We now clarify this in the Methods section as follows:

“cDNA from BT827 and BT618 had been exhausted for RNA-seq analysis and was not available to perform qPCR. BT646 contained a complex rearrangement involving multiple breakpoints that linked KIAA1549 to BRAF that was not amenable to this qPCR approach.”

References

1. La Manno, G. *et al.* RNA velocity of single cells. *Nature* **560**, 494-498 (2018).
2. Qiu, X. *et al.* Reversed graph embedding resolves complex single-cell trajectories. *Nat Methods* **14**, 979-982 (2017).
3. Sun, Y. *et al.* A brain-penetrant RAF dimer antagonist for the noncanonical BRAF oncoprotein of pediatric low-grade astrocytomas. *Neuro Oncol* **19**, 774-785 (2017).
4. Meissner, A. *et al.* Genome-scale DNA methylation maps of pluripotent and differentiated cells. *Nature* **454**, 766-70 (2008).
5. Verhaak, R.G. *et al.* Integrated genomic analysis identifies clinically relevant subtypes of glioblastoma characterized by abnormalities in PDGFRA, IDH1, EGFR, and NF1. *Cancer Cell* **17**, 98-110 (2010).
6. Collins, V.P., Jones, D.T. & Giannini, C. Pilocytic astrocytoma: pathology, molecular mechanisms and markers. *Acta Neuropathol* **129**, 775-88 (2015).
7. Fangusaro, J.R. *et al.* A phase II prospective study of selumetinib in children with recurrent or refractory low-grade glioma (LGG): A Pediatric Brain Tumor Consortium (PBTC) study. *J Clin Oncol* **35**(2018).
8. Jacob, K. *et al.* Genetic aberrations leading to MAPK pathway activation mediate oncogene-induced senescence in sporadic pilocytic astrocytomas. *Clin Cancer Res* **17**, 4650-60 (2011).
9. Raabe, E.H. *et al.* BRAF activation induces transformation and then senescence in human neural stem cells: a pilocytic astrocytoma model. *Clin Cancer Res* **17**, 3590-9 (2011).
10. Kamminga, L.M. *et al.* The Polycomb group gene *Ezh2* prevents hematopoietic stem cell exhaustion. *Blood* **107**, 2170-9 (2006).
11. Whitfield, M.L. *et al.* Identification of genes periodically expressed in the human cell cycle and their expression in tumors. *Mol Biol Cell* **13**, 1977-2000 (2002).
12. Bandopadhyay, P. *et al.* Long-term outcome of 4,040 children diagnosed with pediatric low-grade gliomas: an analysis of the Surveillance Epidemiology and End Results (SEER) database. *Pediatr Blood Cancer* **61**, 1173-9 (2014).
13. Krishnatry, R. *et al.* Clinical and treatment factors determining long-term outcomes for adult survivors of childhood low-grade glioma: A population-based study. *Cancer* **122**, 1261-9 (2016).

REVIEWERS' COMMENTS:

Reviewer #1 (Remarks to the Author):

The authors have addressed all of my concerns with the original manuscript.

Reviewer #2 (Remarks to the Author):

The revision has significantly improved the novelty and impact of the manuscript. My concerns were adequately addressed.

Mitogenic and progenitor gene programs in single pilocytic astrocytoma cells
Reitman et al.
Nature Communications

Reply to Reviewers' Comments

REVIEWERS' COMMENTS:

Reviewer #1 (Remarks to the Author):

The authors have addressed all of my concerns with the original manuscript.

We thank the Reviewer for their feedback on our manuscript.

Reviewer #2 (Remarks to the Author):

The revision has significantly improved the novelty and impact of the manuscript. My concerns were adequately addressed.

We thank the Reviewer for their feedback on our manuscript.